ecology, evolution

movement ecology, adhesive locomotion, optimal foraging, behaviour, sea turtle, *Chelonibia testudinaria*

**Author for correspondence:**
Benny K. K. Chan
e-mail: chankk@gate.sinica.edu.tw

†These authors contributed equally to this study.

# Five hundred million years to mobility: directed locomotion and its ecological function in a turtle barnacle

Benny K. K. Chan[1,†], Yue Him Wong[2,†], Nathan J. Robinson[3,†], Jr-Chi Lin[1], Sing-Pei Yu[1], Niklas Dreyer[1,4,5,6], I-Jiung Cheng[8], Jens T. Høeg[7] and John D. Zardus[9]

[1]Biodiversity Research Center, Academia Sinica, Taipei, Taiwan
[2]Institute for Advanced Study, Shenzhen University, Shenzhen, China
[3]Fundación Oceanogràfic, Oceanogràfic de València, Valencia, Spain
[4]Taiwan International Graduate Program, TIGP, Biodiversity, Academia Sinica, Taipei, Taiwan
[5]Department of Life Sciences, National Taiwan Normal University, Taipei, Taiwan
[6]Natural History Museum of Denmark, and [7]Marine Biology Section, Department of Biology, University of Copenhagen, Copenhagen, Denmark
[8]Institute of Marine Biology, National Taiwan Ocean University, Keelung, Taiwan
[9]The Citadel, Department of Biology, Charleston, SC, USA

BKKC, 0000-0001-9479-024X; NJR, 0000-0001-7384-3576; ND, 0000-0002-1391-1642; JH, 0000-0002-3596-1691

Movement is a fundamental characteristic of life, yet some invertebrate taxa, such as barnacles, permanently affix to a substratum as adults. Adult barnacles became 'sessile' over 500 Ma; however, we confirm that the epizoic sea turtle barnacle, *Chelonibia testudinaria*, has evolved the capacity for self-directed locomotion as adults. We also assess how these movements are affected by water currents and the distance between conspecifics. Finally, we microscopically examine the barnacle cement. *Chelonibia testudinaria* moved distances up to 78.6 mm yr$^{-1}$ on loggerhead and green sea turtle hosts. Movements on live hosts and on acrylic panels occasionally involved abrupt course alterations of up to 90°. Our findings showed that barnacles tended to move directly against water flow and independent of nearby conspecifics. This suggests that these movements are not passively driven by external forces and instead are behaviourally directed. In addition, it indicates that these movements function primarily to facilitate feeding, not reproduction. While the mechanism enabling movement remained elusive, we observed that trails of cement bore signs of multi-layered, episodic secretion. We speculate that proximal causes of movement involve one or a combination of rapid shell growth, cement secretion coordinated with basal membrane lifting, and directed contraction of basal perimeter muscles.

> … the earth, the very emblem of solidity, has moved beneath our feet like a thin crust over a fluid … —Charles R. Darwin, *The Voyage of the Beagle* 1839 [1, p. 217]

## 1. Introduction

The goal of movement ecology is to determine how, why, where and when organisms move [2–4]. Answering these questions can be challenging for organisms with complex life histories involving varying mobility [5]. For example, almost all barnacles progress from swimming planktonic larval stages to a sessile benthic adult stage [6]. Once affixed to a substratum by their adhesive cement [7–11], barnacles are generally considered incapable of further lateral movement unless driven by asymmetric growth, shell repair or reattachment [12–17]. In striking

contrast, the epizoic barnacle *Chelonibia testudinaria* (Linnaeus, 1758) has been reported translocating distances greater than 30 cm over several months' time across sea turtle carapaces [18]. This singular report is particularly notable considering that active locomotion has not been observed in any other barnacle species since their origin 500 Ma [19] and that the underlying mechanisms and adaptive purposes of such movements are currently unknown. The substratum inhabited may bear on the phenomenon, but while some epibiotic barnacles associate with particular sea turtle species [20], *C. testudinaria* is the least host-specific of the turtle barnacles [21] and is also found on manatees [22] and crabs [23].

The most parsimonious explanation for lateral movements in *C. testudinaria* is displacement by external forces, such as water currents. However, as determined previously, barnacles move predominantly towards the head of the turtle and thus into the prevailing flow [18]. Alternatively, turtles could also actively 'knock' barnacles in a certain direction when scraping their carapaces on reefs or rock ledges [24]. Yet such behaviour is unlikely to lead to continual movements over time. Consideration must therefore be given to the speculation that *C. testudinaria* is capable of intrinsically initiating active locomotion. Like several other acorn barnacles, *C. testudinaria* has a flexible basal membrane, a likely critical pre-adaptation for movement in this species, rather than a rigid calcareous basal plate common to many species. The basis is supplied with cement ducts out of which adhesive cement is expelled throughout the life of the animal [25,26], perhaps modulating periodic attachment and detachment in the case of *C. testudinaria*.

Why (and when) *C. testudinaria* makes directed movements could be to increase the chance of finding a mate. Barnacles are unusual among sessile organisms in reproducing via pseudo-copulation between outcrossing hermaphrodites [27]. This creates considerable selection pressure for individuals to settle in clusters because they are constrained to mating with adjacent neighbours [28], hence their possession of unusually long penises [29]. In addition, *C. testudinaria* has evolved an exceedingly rare androdioecious sexual system wherein complemental (i.e. dwarf) males and hermaphrodites occur within the same reproductive population [30]. This may be a response to reduced mating-group size [31–35] combined with the rarity of this barnacles' substratum. Thus, active locomotion to bring individuals closer together could further increase the probability of successful mating.

An alternate explanation for the ecological role of locomotion in *C. testudinaria* is to improve its feeding position. Barnacles are suspension feeders, employing a fan of cirral appendages to capture organic particles and small organisms [36]. As foraging success is aided by water currents, barnacle larvae generally seek to settle in positions where flow is high [37]. Yet, for *C. testudinaria*, its original settlement location may not remain optimal indefinitely because flow dynamics change as the host grows or alters its behaviour. Additionally, current flow is particularly important for *C. testudinaria* because, as a probable consequence of its epizooic lifestyle, it is the only barnacle known to rely entirely on passive feeding [38]. Thus, active locomotion could help ensure that this barnacle remains in optimal feeding locations on its dynamic, living substratum.

To confirm directed locomotion in *C. testudinaria*, we monitored barnacles on wild and captive sea turtles over several months. We also conducted laboratory experiments using individual barnacles reattached to synthetic substrata to test if movement improved their positioning for feeding or reproduction. Furthermore, we examined the barnacle adhesive cement under light and scanning electron microscopy (SEM).

## 2. Material and methods

### (a) Barnacle movements on captive and wild turtles

Loggerhead turtles were caught incidentally by local fisheries within the Communidad de Valencia, Spain, and brought for rehabilitation to the Área de recuperación y conservación de fauna marina (ARCA), which is managed by the Fundación Oceanogràfic under permits granted by the Valencian Regional Government. For this experiment, we selected five turtles admitted to the ARCA between December 2019 and May 2020 that each hosted at least 10 *C. testudinaria*. Upon admission to ARCA, the turtles were suffering from decompression sickness and were thus treated and monitored for up to 18 weeks before being released back into the wild (for details on veterinary care, see [39]). Sea turtles were housed in circular tanks, ranging from 2 to 6 m in diameter with a water depth of 0.95 m and maintained at a water temperature of approximately 24°C. All animals were fed twice daily using a mix of vegetable and fish material.

We monitored barnacle movements on the five turtles in the ARCA by epoxying 1 cm scale bars within 30 mm of any clusters of *C. testudinaria* on the turtles' carapaces. Every two weeks, we photographed each cluster of barnacles with the camera placed perpendicular to the carapace and the nearest 1 cm scale bar within the field of view. We measured the growth and movements of 10 randomly selected barnacles from each turtle in IMAGEJ (v. 1.52p) by using the 1 cm scale bars as fixed reference points of known size. To determine the directionality of the barnacle movements, we noted whether the barnacle moved towards the front of the carapace (i.e. anteriorly), towards the back of the carapace (i.e. posteriorly), towards the midline of the carapace (i.e. medially) or away from the midline of the carapace (i.e. laterally). We also noted whether each individual *C. testudinaria* bore attached complemental males.

To obtain observations of barnacle movements in the wild, we asked professional underwater divers at Siao Liu Qiu Island, Taiwan, for photographs of turtles with barnacles. We received images of three green turtles that both had suitable numbers of barnacles and were repeatedly photographed over a 16-week period. We used the shape of the post-ocular scutes to confirm the identity of the turtles and then inspected the photos visually to determine the movements of the barnacles [40].

### (b) Barnacle cementation and translocation on synthetic substrata

We collected *Charybdis* crabs from the shores of Taiwan using crab traps and selected individuals with *C. testudinaria* barnacles (electronic supplementary material, figure S2A). To remove the barnacles, the crabs were euthanized and each crab's carapace was carefully trimmed to the edge of each barnacle's base. Within 3–4 days, the remaining carapace adhering to the barnacles' bases degraded (approx. 3–4 days) without damaging the membranous base of the barnacles. The 'cleaned' barnacles were placed on 15 × 15 cm acyclic plates until reattached (electronic supplementary material, figure S2B). Successful reattachment of the barnacles was confirmed by the appearance of white cement around the periphery of the base (electronic supplementary material, figure S2C). In total, 15 specimens successfully reattached.

We monitored the movements of these 15 *C. testudinaria* using time-series photographs for up to 1 year in a polyethylene

tray (70 × 20 × 10 cm) with continuous aeration. Each barnacle was photographed once per week at the apical, lateral and basal view using a digital camera (Panasonic, Lumix DMC-G1). On each acrylic plate, four yellow marker dots were placed on the plate as reference points and empty shells of *C. testudinaria* were attached to the plate as position calibration markers. The specimens were fed live adult *Artemia* once per day, the seawater was changed daily and the shell surfaces were brushed clean every 3 days to avoid algal overgrowth.

This experiment was repeated but in the second instance, we used three *C. testudinaria* obtained from the carapaces of dead stranded green sea turtles. These barnacles were once again photographed in apical, lateral and basal view but photos were taken daily instead of weekly. These barnacles were monitored for three, five and eight months, respectively.

## (c) Barnacle cement trail analysis

Observing that moving barnacles often left behind a trail of cement, we inspected the cement to search for other insights into the mechanism enabling barnacle locomotion. We discovered that cement readily detached from its substrate after air-drying for more than one week. We attempted further air-drying (up to a month) but this resulted in cracking of the material (figure 3a). We also attempted to use SEM (FEI Quanta 200) to obtain cross-section views of the internal microstructure of the cement trail; however, cutting the cement with a razor blade destroyed the microstructure along the cut edge (data not shown). Instead, we examined the edges of natural cracks that ran almost perpendicular to the longitudinal axis of the cement strip. Cement trail fragments were mounted on SEM stubs and gold coated prior to observation.

## (d) Barnacle detachment force

To compare the attachment strength of barnacles reattached to acrylic plates and those naturally adhered to crab carapaces, we applied a shear force parallel to the substratum following methods specified in ASTM (2005) [41]. The force required to detach the barnacle was measured with a force gauge (FG-20G, Taiwan). Barnacles on acrylic plates were divided into four groups for analysis based on their reattachment time and nested by size (1–10 ($n = 10$), 11–20 ($n = 5$), 21–30 ($n = 10$) and greater than 365 days ($n = 5$)). Individuals from the final cohort were similar in size to those naturally attached to crabs to which they were compared. Variation in the force needed to detach the barnacles was analysed using one-way analysis of variance (ANOVA) after the data passed a homogeneity of variance test [42].

## (e) Barnacle locomotion relative to current flow

To study the effect of unidirectional water flow on barnacles, six individuals were reattached to the centre of replicate 60 × 60 mm acrylic plates. Each plate was placed in a 60 × 50 × 28 cm tank and positioned in the flow of an underwater pump (flow rate 10 l min$^{-1}$) directed out of a cylindrical PVC funnel that was made by cutting the bottom out of a 250 ml commercial PVC bottle. The internal diameter of the outflow tube of the underwater pump was 9.24 mm and, thus, we estimated the flow velocity to be 2.48 m s$^{-1}$ (velocity = flow rate × tube cross-section area). The experimental design included six replicate plates for three different treatments: (i) barnacles with the rostrum (cirral net) facing towards the flow (i.e. facing the narrow opening of the funnel), (ii) barnacles with the rostrum (cirral net) facing away from the flow (i.e. facing the wide opening of the funnel), and (iii) barnacles on control plates inserted into the funnel but with the water pump shut off. Water temperatures were maintained at 25°C for all experiments (electronic supplementary material, figure S1A–C). The distances traversed

and the angles of movement of barnacles were measured after three months (electronic supplementary material, figure S1D), using the intersection point of the paired scutum and tergum opercular plates of the barnacle as the centre point. All barnacles were fed with *Artemia* nauplii, dispersed and recirculated for 6 h d$^{-1}$. After feeding, the tank water was replaced.

## (f) Barnacle locomotion relative to conspecifics

To investigate whether barnacles move to reduce mating distance, we attached six pairs of barnacles on 16 × 16 cm acrylic plates at inter-individual distances of 5 and 10 cm, and another six pairs of barnacles on 16 × 24 cm acrylic plates at inter-individual distances of 15 cm. The plates were maintained in 25°C aerated seawater. Barnacles were allowed to reattach for three weeks prior to taking any measurements. Barnacles were fed 6 h d$^{-1}$ and the experiment was monitored for 10 months. We analysed variation in inter-individual distances in each treatment using paired *t*-tests.

For a second experiment, we placed five barnacles on to 16 × 16 cm acrylic plates at an inter-individual distance of 5 cm (electronic supplementary material, figure S1E). We ran five replicates for this experiment and each experiment lasted for three months. To assess whether barnacles were clustering, separating or moving randomly, we calculated the area of the minimum convex polygon needed to encompass the area occupied by the barnacles both at the start and end of the experiment. The differences in areas were analysed using paired *t*-tests (figure 4).

For a third experiment, we reattached 29 and 31 individuals onto separate 36 × 36 cm acrylic plates. The barnacles were placed at inter-neighbour distances ranging between 30 and 50 mm. Photographs of barnacles were collected weekly over 12 weeks. In each photograph, the distance to the nearest neighbour for each barnacle was measured using image analysis software (SIGMA SCAN PRO 5) and a nearest neighbour index (NNI) was used to assess the distribution [43] using the following formula:

$$R_n = \frac{\bar{D}(\text{Obs})}{0.5 \sqrt{a/n}},$$

where '$R_n$' is the NNI value, '$\bar{D}(\text{Obs})$' is the average observed nearest neighbour distance in cm between the barnacles, '$a$' is the spatial area in cm$^2$ occupied by the barnacles and '$n$' is the total number of individual barnacles. Values of the $R_n$ index range in a continuum from 0 to 2.15. $R_n$ values close to 0 indicate a clustered distribution, a value of 1 indicates a random distribution and a value near 2.15 indicates a regular distribution.

During all laboratory movement experiments, barnacles were kept at 25°C water temperature, which is typical of seawater temperatures during summer in Taiwan when most barnacles produce mature gonads for reproduction. Several *C. testudinaria* had mature ovaries and testes upon dissection at the end of the experiment, suggesting that many individuals were reproductively active during the experiments.

# 3. Results

## (a) Translocation on live turtles

We monitored the movement and growth of 50 *C. testudinaria*, 10 on each of five captive loggerhead sea turtles (*Caretta caretta*) over 14–18 weeks. The shell diameter of the barnacles (maximum rostro-carinal length) initially ranged from 14 to 41 mm (mean 24 mm) and increased to 14 to 45 mm (mean 26 mm) by the end of the study (electronic supplementary material, figure S4). To account for any effect of barnacle growth on the movement measurements, we concluded translocation had occurred only when the distance travelled

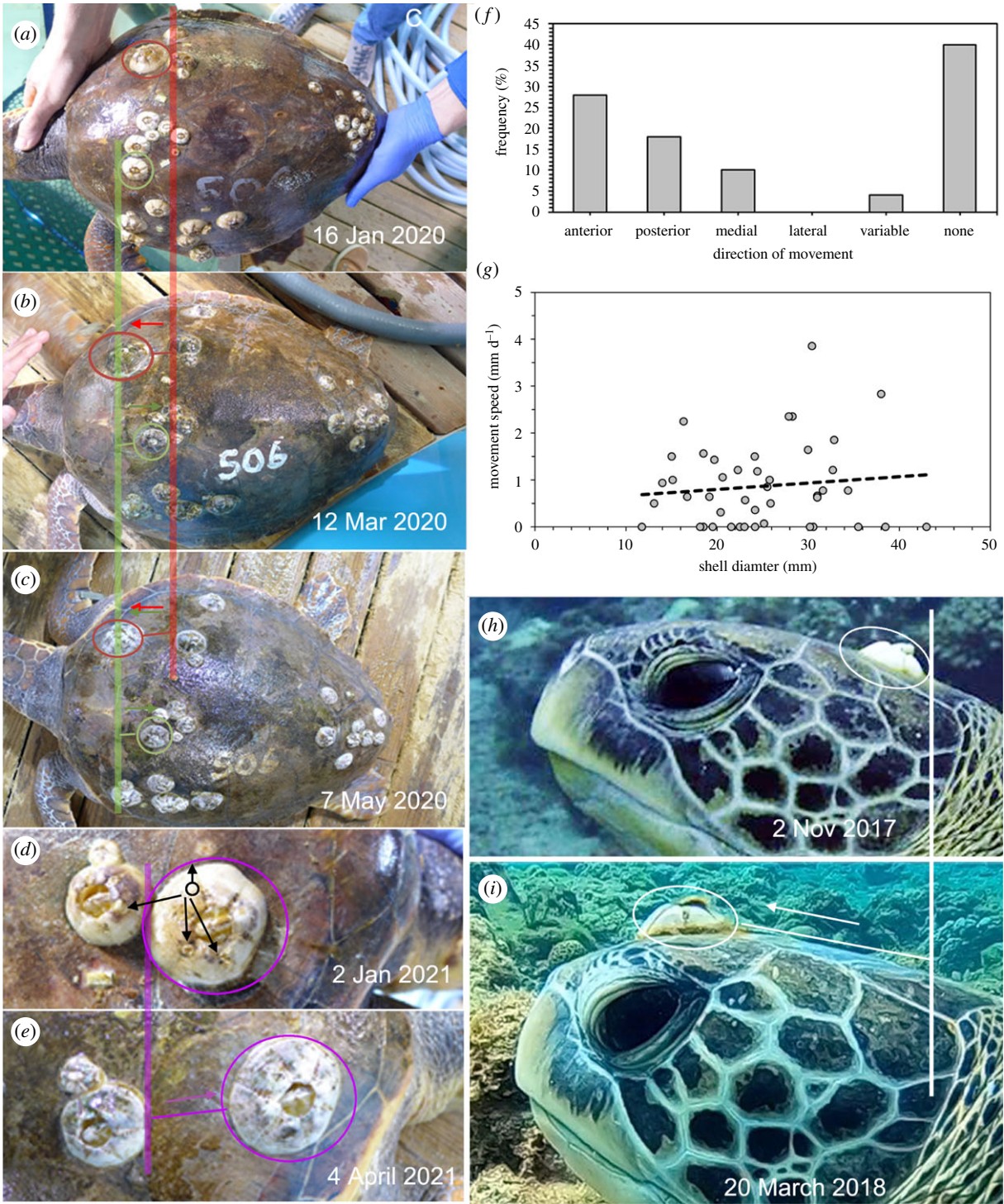

**Figure 1.** Growth and movement of barnacles on captive loggerhead and wild green sea turtles. (*a*–*c*) Five loggerhead turtles were photographed every two weeks for up to 18 weeks. Red and green circles highlight the starting position of two barnacles that moved distances over 30 mm on a single turtle (curved carapace length: 52 cm) within the study period. (*d,e*) Movement of two barnacles with attached complemental males (black arrows) on loggerhead turtles. (*f*) Direction travelled by 30 barnacles on the carapaces of loggerhead turtles. Variable direction refers to individuals that changed direction during the study. (*g*) Variation in moving speed and barnacle size on captive loggerhead turtles. (*h,i*) Movement of a barnacle on the head of a wild green sea turtle over 138 days. (Online version in colour.)

was greater than 5 mm in 14 weeks, as this exceeded the fastest growth rate observed in any barnacle in the study. A total of 30 barnacles were documented moving at a speed of 1.4 mm per week including three individuals with complemental males (figure 1*a*–*e*). The farthest distance any moved was 54 mm (0.5 mm d$^{-1}$). We observed that 14 barnacles travelled towards the front of the carapace (i.e. anteriorly), nine travelled to the back to the carapace (i.e. posteriorly), five travelled towards the midline of the carapace (i.e. medially), while no barnacles

moved away from the midline of the carapace (i.e. laterally) (figure 1*f*). Two barnacles changed direction midway through the study: one barnacle initially moved in a posterior direction before moving anteriorly, while another initially moved in an anterior direction before veering medially. Movement speed was not significantly correlated with shell diameter (Pearson's correlation coefficient test: $R^2 = 0.11$, $n = 44$, $p = 0.47$) (figure 1*g*).

We also monitored barnacle movements from opportunistic time-series photos of three wild green sea turtles *Chelonia*

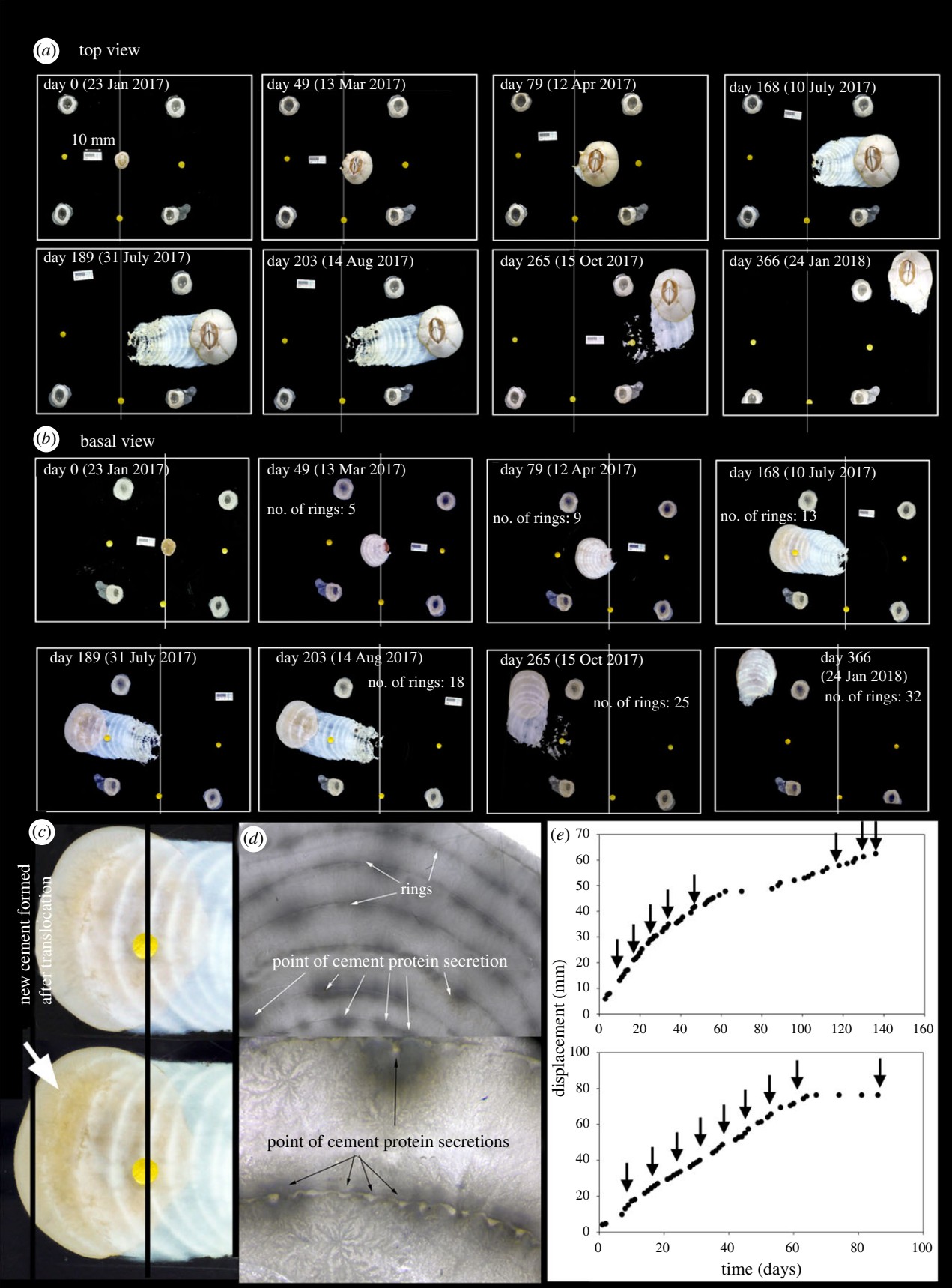

**Figure 2.** Growth and movement of barnacles on synthetic substratum (no flow, aeration only) over 1 year following reattachment to transparent acrylic panels. Lines, yellow dots and empty shells serve as spatial reference points. (*a*) Topside view of a panel. Time-series photos of a barnacle 'surfing' on a layer of cement. By day 79, notable movement had commenced, leaving a trail of cement. On day 203, the barnacle changed direction 90° to the left with no associated shell rotation. (*b*) Underside view of the same panel showing basal views of the cement trail. (*c*) Basal view of the barnacle on day 168 (upper image) and day 175 (lower image). At day 168, there was no apparent secretion of new cement. By day 175, a faint rim of fresh cement was observed lagging the leading edge of the shell. (*d*) Cement trail viewed under a stereomicroscope with incident light source (lower view at high magnification), showing rings of cement and points of secretion along the annuli. (*e*) Displacement of two selected barnacles (upper and lower plots, respectively) transferred from sea turtles onto panels based on daily observation of bases. Arrows indicate the formation of asymmetrical rings. (Online version in colour.)

*mydas* from Siao Liu Qiu Island, Taiwan. Barnacles were observed on the heads of two separate turtles and one of these turtles had a barnacle positioned along a marginal scute of the carapace (figure 1*h,i*). When the same turtles were photographed over 16 weeks later, all barnacles had moved in an anterior direction on the turtles. A trail of white cement was observed marking the movement pathway (electronic supplementary material, figure S5).

## (b) Locomotion on synthetic substrata

We removed 15 *C. testudinaria* from *Charybdis* crabs (electronic supplementary material, figure S2A,B) and relocated them onto transparent acrylic panels in the laboratory. Within three months, 14 of the reattached individuals (93%) had moved a mean distance of 7 mm (0.08 mm d$^{-1}$; electronic supplementary material, figure S3). Most individuals moved in a single direction, but not always rostrum-first. One individual during the trial altered course by 90° at 265 days post reattachment yet did not alter the orientation of its body. In total, this individual moved 79 mm in 365 days (0.22 mm d$^{-1}$; figure 2*a,b*). While moving, all barnacles left behind a trail of secreted cement (figure 2*a,b*).

In a subsequent relocation experiment, we used three *C. testudinaria* collected from the carapaces of dead stranded green sea turtles and the movement of each barnacle was monitored using time-series photography at daily intervals for up to three, five and eight months, respectively. The daily photographs indicated barnacles moved intermittently, forming a ring of cement at temporary 'stops' (figure 2*c,d*). Rings were made every approximately 8–10 days and the distance between them was approximately 10–14 mm (figure 2*e*). In between these rings, movement occurred at 0.5–0.9 mm d$^{-1}$ (figure 2*e*).

## (c) Adhesive strength of cement

We measured the force required to remove 30 barnacles from their substratum at several time intervals (1–10, 11–20, 21–30, greater than 365 days) post reattachment. Barnacle size was positively correlated with attachment strength ($F_{4,30} = 4197$, $p < 0.05$), probably owing to the increased surface area available for adhesion. *Post hoc* Student–Newman–Keuls tests revealed that by 365 days there was no significant difference in the force needed to remove individuals on synthetic panels (approx. 42 N) compared to those naturally occurring on crabs (approx. 48 N) (electronic supplementary material, figure S6).

## (d) Cement secretion

Observation of the cement trails by light microscopy showed that they are secreted at the leading edge of movement (figure 2*c*). In addition, after several months, many individuals had deposited a trail of overlapping rings of cement indicating discrete episodes of secretion (figure 2*c,d*). The number of rings was correlated with distance (figure 2*e*) and they changed orientation in tandem with movement of the individuals (figure 2*a*). Further investigation of the cement trails under SEM (figure 3*a–e*) revealed microstructural variations that coincided with position (time since secretion). When viewed on edge from the side, newer cement appeared to be a homogeneous solid composed of cubic crystals that originated from below adjacent, older deposits which appeared heterogeneous (figure 3*b,c*), while each instalment exhibited

an alternating bi-layered construction indicating some discontinuity during secretion (figure 3). Appearing in regular, alternating arrangement (figure 3*f*), the outer layer looked similar to secondary cement seen in barnacles that have partially lifted and reattached [15]. Such cement trails further support the conclusion that secretion and movement is incremental and not continuous (figure 3*f*).

## (e) Does locomotion improve foraging conditions?

Barnacles reattached to acrylic sheets with their cirral nets facing towards the current moved mean distances more than four times farther than barnacles in other treatments ($F_{2,15} = 4.88$, $p = 0.023$; figure 4*a*). Those barnacles with cirral nets facing towards the current tended to travel forwards (average 327° angular travel, figure 4*b*; note 0/360° is the direction from where the current originated), while barnacles with cirral nets facing away from the current moved backwards (average 178° angular travel, figure 4*b*). All barnacles in the random control treatment moved distances less than 0.3 mm and did not move in a directed manner (figure 4*a,b*).

## (f) Does locomotion increase reproductive opportunity?

The movements of barnacles placed on acrylic plates at three inter-individual treatment distances (50, 100 and 150 mm) did not significantly decrease inter-neighbour proximity (paired *t*-test, $p > 0.05$; figure 4*c*). By contrast, individuals moved in all directions, with most individuals repositioning themselves at angles of 80–240° (figure 4*d*). Similarly, there was no statistical change over time in the size of the minimum convex polygon for barnacles that were reattached in groups of five onto single acrylic sheets (paired *t*-test, $p > 0.05$; figure 4*e*). Finally, barnacles were reattached in clusters of 29 and 31 individuals on two separate acrylic sheets. The movements of these individuals over 12 weeks were analysed using nearest neighbour analysis and the NNI of these panels changed from 1.3 to 0.9 and 1.4 to 1.2, respectively (figure 4*f,g*). As NNI values near 1 indicate a random distribution pattern, values near zero signify clustering, and values approaching 2.15 connote an even distribution, this indicates that the distribution of the barnacles became increasingly random over time.

## 4. Discussion

Our results unequivocally confirm the observations of Moriarty *et al.* [18] that *C. testudinaria* is capable of directed movement, contradicting the common belief that all adult barnacles are inherently sessile. Our field observations and experiments indicate that these movements serve primarily to move individuals into areas of higher current flow. This suggests that barnacles use locomotion to improve feeding conditions instead of to enhance reproductive opportunity, though both outcomes are not necessarily mutually exclusive. The mechanism enabling these animals to translocate requires further study, but the ecological impact of this adaptation for *C. testudinaria* is undoubtedly profound. While it seems unlikely that acorn barnacles possessing a rigid calcareous base are capable of directed locomotion, the phenomenon of active translocation merits investigation in other barnacle species from sea turtles [21,44], species with membranous bases in general and other taxa that are sessile. Indeed, a recent study has demonstrated that a species of deep-sea sponge is also unexpectedly capable of

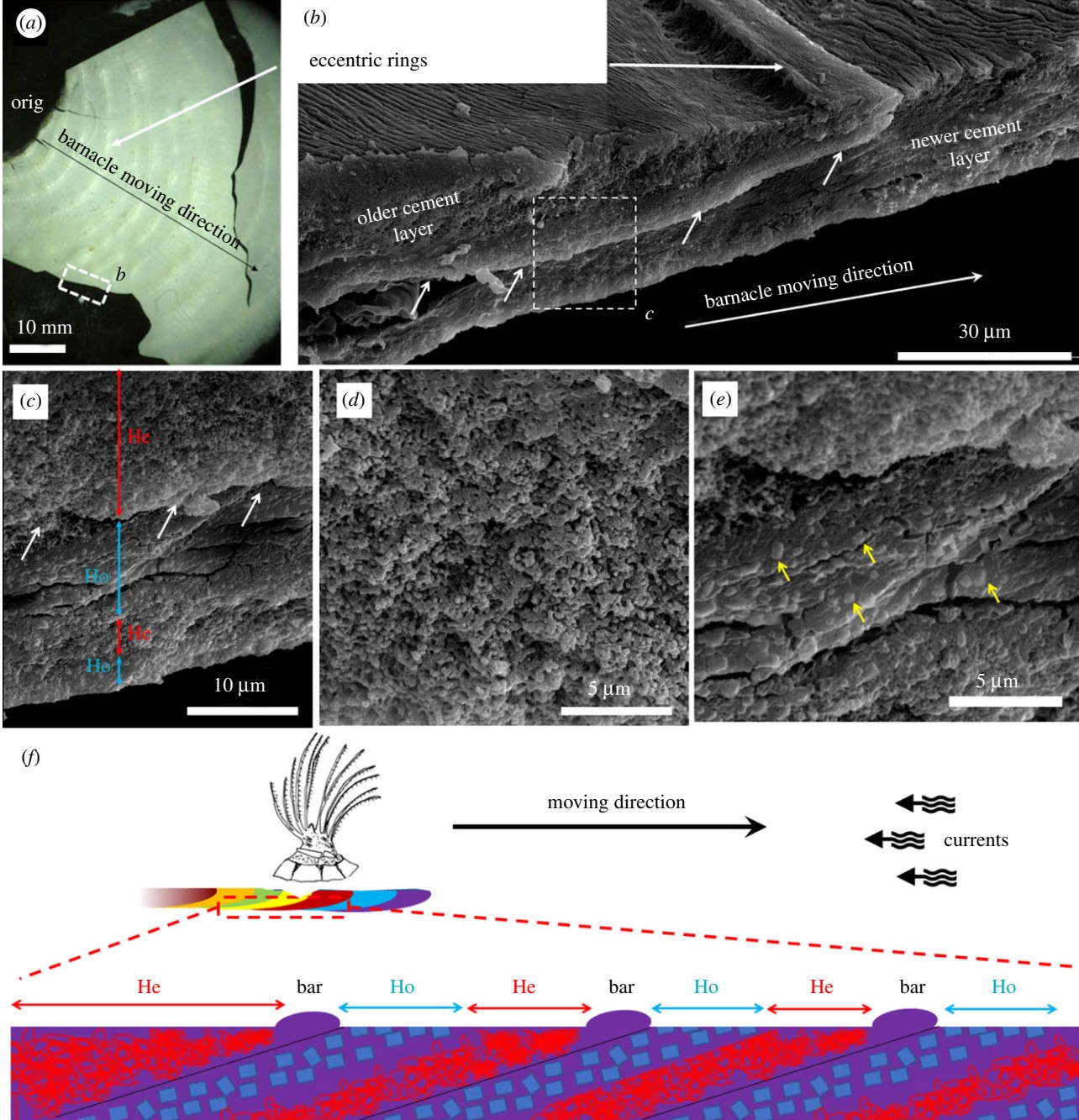

**Figure 3.** Imaging of cement plagues. (*a*) Piece of a torn cement plaque (orig: initial reattachment point; black arrow: direction of movement). (*b*) Edge-view by scanning electron microscope (SEM) of the torn cement plaque from (*a*) (small, white dotted box; small white arrows indicate the boundary between two independent secretion events). (*c*) Detail from (*b*) (white dotted box) showing rough heterogeneous cement (He, in red) and smooth homogeneous cement (Ho, in blue). (*d*) High magnification of heterogeneous cement. (*e*) High magnification of homogeneous cement embedded with small cubic crystals. (*f*) Side view diagram illustrating micro-stratification of alternating cement bi-layers in the cement trail. Elevated bars correspond to the concentric rings seen in (*a*) and mark the end of a secretion episode. (Online version in colour.)

spatial translocation [45], providing further incentive to study the movement ecology of other ostensibly sessile invertebrates.

## (a) How do turtle barnacles move?

Barnacles under laboratory conditions and in the wild both showed a clear tendency of moving into the prevailing current and so these movements cannot be the product of hydrodynamic forcing. Moreover, adhesion for all individuals at any time point was sufficiently strong to rule out water flow as a potential motive force. This strongly supports the idea that the observed barnacle movements represent directed locomotion, yet perhaps the most convincing evidence is that we recorded four instances where barnacles

shifted their direction abruptly mid-experiment without any perturbation to the experimental set-up (figure 2*a*).

For barnacles to be capable of active movement, they must periodically adhere and detach from a substratum. Barnacles adhere via a cement formed of highly hydroxylated soluble and insoluble proteins [10,11]. *Chelonibia testudinaria* must, therefore, be able to periodically dissolve the cement or detach from it during bouts of movement before reattaching. We observed that *C. testudinaria* secretes droplets of cement to form distinct layers that probably represent sequential secretion events. This contrasts with common acorn barnacles (family Balanidae), whose cement secretions form a thin monolayer of adhesive made from nanometre-sized globules that cure as a gel-to-solid on

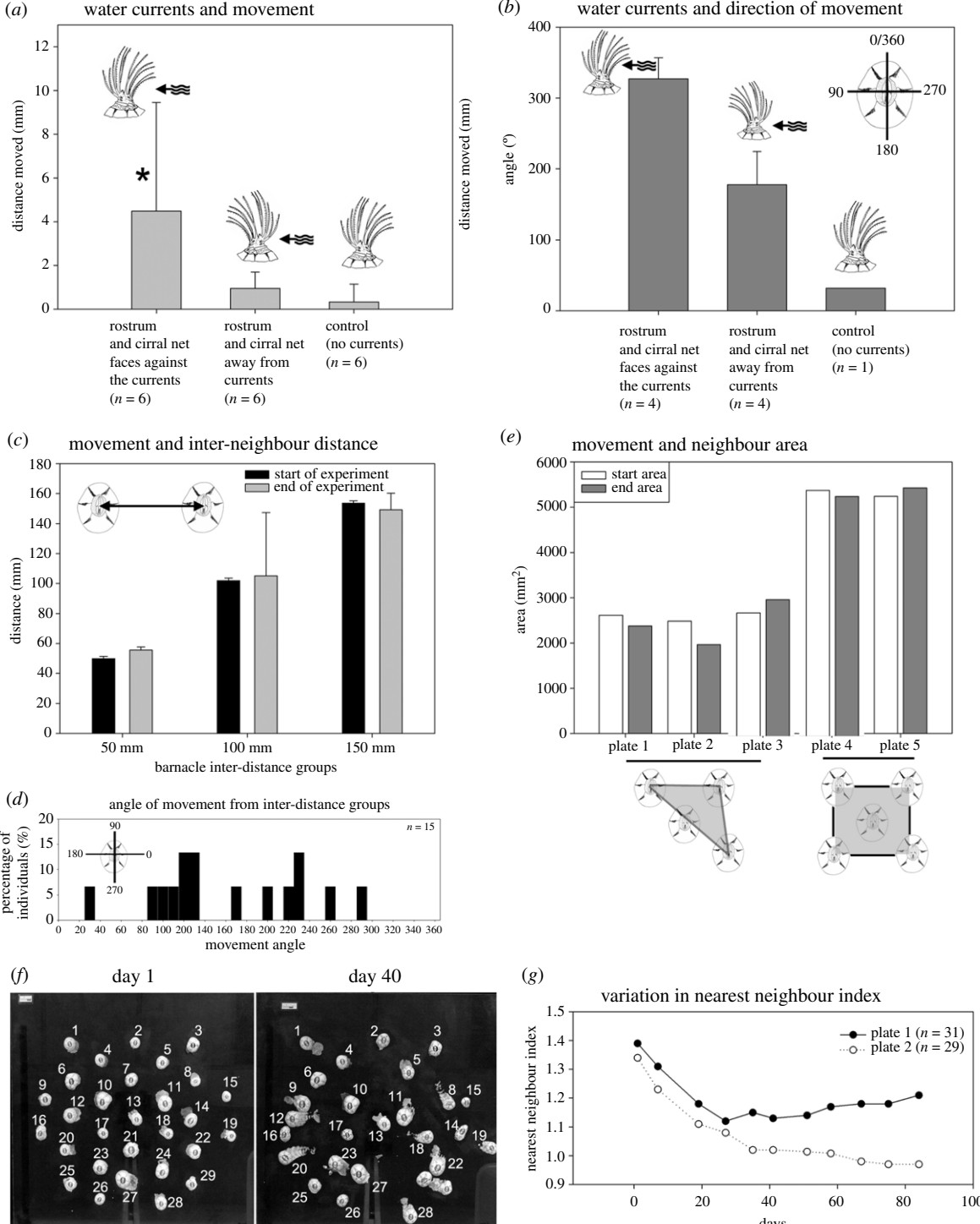

**Figure 4.** Manipulated water current (see the figure for methods) and nearest neighbour-distance experiments. (*a*) Mean (±1 s.d.) distances barnacles moved when oriented with rostrum and cirral net facing towards the current, rostrum and cirral net facing away from current, and control (no current). (*b*) Mean (+1 s.d.) angular declination of barnacle movement relative to unidirectional flow (0/360°), with barnacle rostrum and cirral net facing towards the current, rostrum and cirral net facing away from the current, and control (no current). Note only the translocating barnacles in each treatment were analysed (*n* = 4 in both flow treatments; *n* = 1 in no-flow control). 0° indicates the anterior position. (*c*) Variation in mean (±1 s.d.) change in inter-individual distances between barnacle pairs over three months in 50, 10 and 15 mm distance-treatments. (*d*) Histogram showing the angle of movement of the inter-distance groups (see the electronic supplementary material, figure S1D for methods). 0° indicates the anterior position. (*e*) Variation in the area of barnacle groups (each with inter-individual distance 50 mm) before and after the experiment (three months). All plates with five barnacles initially attached, but one barnacle died in plates 1–3 within two weeks of set-up. (*f*) Plate with 29 regularly spaced barnacles attached (each with 3–5 mm inter-neighbour distance) at day 1 and at day 40. (*g*) Variation in nearest neighbour values of barnacles on two acrylic plates over a period of 84 days. Note, a nearest neighbour value close to 1 indicates a random distribution.

the substratum [46,47]. Another factor undoubtedly important to movement is that *C. testudinaria* possesses an uncalcified, membranous base. This means that since new cement is secreted near the periphery of the basal membrane, the central region, if detached from the substratum, does not easily re-adhere. This

circumstance may facilitate translocation by reducing the area needing to be periodically untethered from the substratum.

Based on our observations of the bi-layered cement and circumferential attachment of *C. testudinaria*, we speculate that this species can periodically release the edge of its basal

membrane before expanding and secreting a new layer of cement into the submembrane space. Using one or a combination of (i) rapid shell growth, (ii) metachronal flexing of either the margin or the entire basal membrane, and (iii) directed internal muscular tension, the barnacle takes advantage of still viscous new cement to slide or 'surf' on the top of the semi-solid adhesive. Small muscles inserting obliquely from the base could power waves in the membrane surface which, coupled with properties of the membrane and adhesive cement, result in progressive movement like pedal waves in the foot of gastropods on mucus during adhesive locomotion [48]. Alternatively, these muscles could contract at the posterior (carinal) margin and somehow 'pull' the whole array of shell plates forward.

## (b) Why do turtle barnacles move?

The capacity of *C. testudinaria* for active locomotion could lead to a range of adaptive possibilities not available to other barnacle species. Our data suggest spatial translocation in *C. testudinaria* serves to optimize foraging conditions. In support of this conclusion, barnacles in experiments on live and synthetic substrata tended to move into the current and when kept in aerated tanks without unidirectional current flow in the laboratory, they moved in random directions, sometimes forward or sideways. By contrast, our laboratory experiments showed that barnacles did not consistently move to reduce inter-neighbour distances and barnacles with complemental males still moved even though they had a potential mate in their immediate proximity. While this suggests that *C. testudinaria* does not necessarily move to increase the probability of hermaphrodite–hermaphrodite copulation, if all individuals move towards areas with higher current to feed, this convergence could simultaneously benefit reproduction as well.

## (c) Where and when do turtle barnacles move?

The most common direction for barnacles to translocate was towards the anterior regions of the turtles' carapaces, consistent with the hypothesis that this enhances feeding. Though clearly rheotactic, barnacles still moved in the absence of flow, but in random directions, possibly in search of feeding currents. The requirement for passive feeding in *C. testudinaria* [43] suggests that movement is a less costly alternative evolutionarily to staying fixed in a location where active feeding is required. Lacking information on energetics, it would be interesting to know what trade-offs exist between flow velocity and locomotion.

In contrast with directionality, there was no correlation between degree of movement and barnacle size (shell diameter),

suggesting movement is not specific to a particular age or time of life of a barnacle. This is further evidence that movement does not occur or increase when individuals are in reproductive phase.

**Ethics.** All vertebrate research was approved by the animal welfare committee of the Fundación Oceanogràfic.

**Data accessibility.** All the data presented in this manuscript are included in the associated figures or electronic supplementary material. Raw data are also available from the Dryad Digital Repository: https://doi.org/10.5061/dryad.crjdfn34d [49].

**Authors' contributions.** B.K.K.C.: conceptualization, data curation, formal analysis, funding acquisition, investigation, methodology, project administration, resources, supervision, validation, visualization, writing—original draft, writing—review and editing; Y.H.W.: conceptualization, data curation, formal analysis, funding acquisition, investigation, methodology, project administration, resources, supervision, validation, visualization, writing—original draft; N.J.R.: conceptualization, data curation, formal analysis, funding acquisition, investigation, methodology, project administration, resources, supervision, validation, visualization, writing—original draft, writing—review and editing; J.-C.L.: conceptualization, data curation, formal analysis, investigation, methodology; S.-P.Y.: methodology; N.D.: conceptualization, writing and editing; I.-J.C.: methodology; J.T.H.: conceptualization, data curation, formal analysis, funding acquisition, investigation, methodology, project administration, resources, supervision, validation, visualization, writing—original draft, writing—review and editing; J.D.Z.: conceptualization, formal analysis, methodology, supervision, visualization, writing—original draft, writing—review and editing. All authors gave final approval for publication and agreed to be held accountable for the work performed therein.

**Competing interests.** The authors declare no competing interests include paid employment or consultancy, stock ownership, patent applications, personal relationships with relevant individuals, and membership of an advisory board.

**Funding.** B.K.K.C. is funded under the research grant in Academia Sinica. Y.H.W. was jointly supported by the Innovation Team Project of Universities in Guangdong Province (no. 2020KCXTD023) and Natural Science Foundation of Shenzhen University award (no. 2019019). Funding was generously provided to J.D.Z. by The Citadel Foundation. N.D. was jointly supported by a double degree graduate grant from the Taiwan International Graduate Program (TIGP) and the Natural History Museum of Denmark (SNM). J.T.H. was financed by the Danish Academy for Independent Research.

**Acknowledgements.** We thank Prof. Shirley Lim (Nanyang Technical University, Singapore) for advice and encouragement on the study and Su-Mei Lin (Academia Sinica), Pei-Chen Tsai, Wei-Peng Hsueh (Academia Sinica) for assisting with laboratory preparations of samples. We thank Jose Luis Crespo Picazo (Fundación Oceanogràfic) and Vicente Marco Cabedo (Fundación Oceanogràfic) for their assistance taking photos of loggerhead turtles in the ARCA. We acknowledge professional divers Su Huai (photo credit for figure 1*h,i*, S5), Da-Bing Zhong, and Chia-ling Feng who supplied underwater photos of sea turtles. The South Carolina Department of Natural Resources assisted in obtaining barnacles from sea turtles in the USA.

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
