## [Peer Review File · Proceedings of the Royal Society B: Biological Sciences]

Review History

RSPB-2021-1257.R0 (Original submission)

Review form: Reviewer 1 (A. Richard Palmer)

Recommendation

Major revision is needed (please make suggestions in comments)

Scientific importance: Is the manuscript an original and important contribution to its field?
Excellent

General interest: Is the paper of sufficient general interest?
Excellent

Quality of the paper: Is the overall quality of the paper suitable?
Good

Is the length of the paper justified?
Yes

Should the paper be seen by a specialist statistical reviewer?
No

Do you have any concerns about statistical analyses in this paper? If so, please specify them explicitly in your report.

No

It is a condition of publication that authors make their supporting data, code and materials available - either as supplementary material or hosted in an external repository. Please rate, if applicable, the supporting data on the following criteria.

Is it accessible?

N/A

Is it clear?

N/A

Is it adequate?

N/A

Do you have any ethical concerns with this paper?

No

Comments to the Author

GENERAL COMMENTS

I) Acknowledgement of prior work on barnacle movement- Contrary to the authors' claims in lines 171-72 (and implications in the abstract and introduction), the present paper does NOT provide the "first unequivocal evidence that the sea turtle barnacle *C. testudinaria* is not permanently sessile". Moriarty et al. (2008) provide compelling photographic evidence that *C. testudinaria*: a) move on the backs of individually tracked green turtles, and b) tend to move upstream.

These earlier results need to be acknowledged more clearly in the abstract, introduction and discussion, and the present authors should explain how their results build upon those of Moriarty et al. 2008.

II) Be consistent throughout the MS to report (in parentheses if preferred) movement rates in mm/day instead of "mm/week" or "X mm in Y weeks".

Also, perhaps comment on the movement rates reported in this MS compared to those reported by Moriarty et al. (2008) of 0.3 - 1.2 mm/d (their Table 1). The movement rates in this study seem to be rather higher (many greater than 1mm/d; Fig. 1G), whereas those on artificial substrata were rather lower (<0.1 mm/d).

SPECIFIC COMMENTS

I have made a number of queries and minor editorial suggestions directly in the annotated PDF of the MS, the more significant ones include (line numbers as in original PDF).

34, 79, 242: Moriarty et al. 2008 have already confirmed that *C. testudinaria* move in a directed way on turtles, so these statements seem a bit misleading. This earlier result should be openly and fully acknowledged. See also General Comment I.

35: "evolved" rather than "revolved"?

38: perhaps mention in the abstract the turtles on which barnacle movement was observed

93-96: The % barnacles moving in given directions in the text does not agree with the numbers in Fig. 1F. For example, the text says "(47 %) headed towards the anterior of the turtles' carapace",

but Fig. 1F indicates only 27-28% did. These discrepancies should be fixed.

155: subhead should be italics, not bold

171-72: This opening statement in the discussion is not correct. Moriarty et al. 2008 provide the first compelling evidence that *C. testudinaria* can move on their turtle host, and that this movement tends to be directional. See also General Comment I.

208-11: An alternative hypothesis might be that small muscles that insert obliquely from the base at the posterior (carinal) margin could contract preferentially, which would tend to 'pull' the whole array of linked lateral plates forward.

271: Revise to read: "barnacle's maximum basal diameter (maximum rostro-carinal length)"

301-3: Comment on approx. water velocities in the aquarium and how barnacles were positioned relative to the direction of flow.

317: Don't you mean Fig. 3A?

327; Fig. S5 legend: State explicitly what the direction of the "detachment force" was: a) shear (parallel to the substratum, as in ASTM 2011 (ref. 36)? or b) perpendicular to the substratum. If the latter, the authors should indicate how they attached the force gauge to the barnacle prior to detachment.

338: Indicate approximate flow velocity. The water velocity for a given volume flow rate depends on the diameter of the opening.

355: Describe in more detail the water flow conditions (approx. velocities, source of water motion)

459: Genus species name italicized

523-24: Explain in Fig. 1 legend whether "direction of movement" means relative to the barnacle body or relative to the turtle carapace (presumably the latter).

528: Mention the flow conditions of the experiment in the title.

539-41: Fig. 2 legend refers to a panel F, but there is no panel F in my copy of Fig. 2.

564: Maybe indicate here "(see Fig. S6 for methods)".

Fig. 1 A-E,H,I: Maybe add arrowheads to the straight lines to indicate the direction of change in barnacle position.

Fig. 1F: See comment regarding text lines 93-96.

Fig. 1G: X-axis label would be more precise as "basal diameter" rather than "shell diameter".

Fig. 3B: Shouldn't the label be "concentric rings" rather than "eccentric rings"?

Fig. 4A,C: Wouldn't it be better to report movement (y-axes) as mm/d to be consistent?

Fig. S4: Increase the font size of the text in the legends of each figure panel so they will be more visible.

Fig. S5 legend: Cite a reference for the SNK test and indicate software used.

Fig. S6 legend: Indicate the approx. flow velocities the barnacles experienced in these treatments.

Review form: Reviewer 2

Recommendation

Major revision is needed (please make suggestions in comments)

Scientific importance: Is the manuscript an original and important contribution to its field?

Acceptable

General interest: Is the paper of sufficient general interest?

Marginal

Quality of the paper: Is the overall quality of the paper suitable?

Acceptable

Is the length of the paper justified?

Yes

Should the paper be seen by a specialist statistical reviewer?

No

Do you have any concerns about statistical analyses in this paper? If so, please specify them explicitly in your report.

No

It is a condition of publication that authors make their supporting data, code and materials available - either as supplementary material or hosted in an external repository. Please rate, if applicable, the supporting data on the following criteria.

Is it accessible?

No

Is it clear?

Yes

Is it adequate?

Yes

Do you have any ethical concerns with this paper?

Yes

Comments to the Author

Dear authors and editor,

Please find my comments and suggestions in the attached pdf document. (See Appendix A)

Decision letter (RSPB-2021-1257.R0)

29-Jun-2021

Dear Dr Robinson:

I am writing to inform you that your manuscript RSPB-2021-1257 entitled "Confirmation of active location and its ecological function by an ostensibly sessile barnacle" has, in its current form, been rejected for publication in Proceedings B.

This action has been taken on the advice of referees, who have recommended that substantial revisions are necessary. With this in mind we would be happy to consider a resubmission, provided the comments of the referees are fully addressed. However please note that this is not a provisional acceptance.

Please note that our data accessibility policy does not allow for raw data to be only available on request to the authors; the raw data are the paper's data and should be provided (e.g. via Dryad) with the revised MS, for publication if the paper is accepted.

Sincerely,
Dr John Hutchinson, Editor
<mailto:proceedingsb@royalsociety.org>

Associate Editor
Comments to Author:

Thank you for the opportunity to review this manuscript, which documents locomotor movements in barnacles that are generally presumed to be sessile, and which conducts tests to evaluate the underlying mechanisms of movement and potential reasons it is undertaken. Reviews from two referees have now been received. These reviews diverged in their regard for the manuscript, but both raised concerns. Referee 1 viewed the findings and consideration of their causes to have significance, but considered the novelty of recognizing barnacle locomotion to be overstated, as a paper that was cited by the authors did, in fact, previously identify such

movements (though its analysis was less extensive). Referee 2 indicated that the Results and Discussion could do more to provide context and explain the significance of the study to a wider audience. Referee 2 also identified several specific issues that require additional clarification.

With these concerns, I cannot recommend this manuscript for publication in its current form. Thank you once again for your submission. I hope you find the referee comments to provide constructive guidance for further work on your study.

Reviewer(s)' Comments to Author:

Referee: 1

Comments to the Author(s)

GENERAL COMMENTS

I) Acknowledgement of prior work on barnacle movement- Contrary to the authors' claims in lines 171-72 (and implications in the abstract and introduction), the present paper does NOT provide the "first unequivocal evidence that the sea turtle barnacle *C. testudinaria* is not permanently sessile". Moriarty et al. (2008) provide compelling photographic evidence that *C. testudinaria*: a) move on the backs of individually tracked green turtles, and b) tend to move upstream.

These earlier results need to be acknowledged more clearly in the abstract, introduction and discussion, and the present authors should explain how their results build upon those of Moriarty et al. 2008.

II) Be consistent throughout the MS to report (in parentheses if preferred) movement rates in mm/day instead of "mm/week" or "X mm in Y weeks".

Also, perhaps comment on the movement rates reported in this MS compared to those reported by Moriarty et al. (2008) of 0.3 - 1.2 mm/d (their Table 1). The movement rates in this study seem to be rather higher (many greater than 1mm/d; Fig. 1G), whereas those on artificial substrata were rather lower (<0.1 mm/d).

SPECIFIC COMMENTS

I have made a number of queries and minor editorial suggestions directly in the annotated PDF of the MS, the more significant ones include (line numbers as in original PDF).

34, 79, 242: Moriarty et al. 2008 have already confirmed that *C. testudinaria* move in a directed way on turtles, so these statements seem a bit misleading. This earlier result should be openly and fully acknowledged. See also General Comment I.

35: "evolved" rather than "revolved"?

38: perhaps mention in the abstract the turtles on which barnacle movement was observed

93-96: The % barnacles moving in given directions in the text does not agree with the numbers in Fig. 1F. For example, the text says "(47 %) headed towards the anterior of the turtles' carapace", but Fig. 1F indicates only 27-28% did. These discrepancies should be fixed.

155: subhead should be italics, not bold

171-72: This opening statement in the discussion is not correct. Moriarty et al. 2008 provide the first compelling evidence that *C. testudinaria* can move on their turtle host, and that this movement tends to be directional. See also General Comment I.

208-11: An alternative hypothesis might be that small muscles that insert obliquely from the base at the posterior (carinal) margin could contract preferentially, which would tend to 'pull' the whole array of linked lateral plates forward.

271: Revise to read: "barnacle's maximum basal diameter (maximum rostro-carinal length)"

301-3: Comment on approx. water velocities in the aquarium and how barnacles were positioned relative to the direction of flow.

317: Don't you mean Fig. 3A?

327; Fig. S5 legend: State explicitly what the direction of the "detachment force" was: a) shear (parallel to the substratum, as in ASTM 2011 (ref. 36)? or b) perpendicular to the substratum. If the latter, the authors should indicate how they attached the force gauge to the barnacle prior to detachment.

338: Indicate approximate flow velocity. The water velocity for a given volume flow rate depends on the diameter of the opening.

355: Describe in more detail the water flow conditions (approx. velocities, source of water motion)

459: Genus species name italicized

523-24: Explain in Fig. 1 legend whether "direction of movement" means relative to the barnacle body or relative to the turtle carapace (presumably the latter).

528: Mention the flow conditions of the experiment in the title.

539-41: Fig. 2 legend refers to a panel F, but there is no panel F in my copy of Fig. 2.

564: Maybe indicate here "(see Fig. S6 for methods)".

Fig. 1 A-E,H,I: Maybe add arrowheads to the straight lines to indicate the direction of change in barnacle position.

Fig. 1F: See comment regarding text lines 93-96.

Fig. 1G: X-axis label would be more precise as "basal diameter" rather than "shell diameter".

Fig. 3B: Shouldn't the label be "concentric rings" rather than "eccentric rings"?

Fig. 4A,C: Wouldn't it be better to report movement (y-axes) as mm/d to be consistent?

Fig. S4: Increase the font size of the text in the legends of each figure panel so they will be more visible.

Fig. S5 legend: Cite a reference for the SNK test and indicate software used.

Fig. S6 legend: Indicate the approx. flow velocities the barnacles experienced in these treatments.

Referee: 2

Comments to the Author(s)

Dear authors and editor,

Please find my comments and suggestions in the attached pdf document.

Author's Response to Decision Letter for (RSPB-2021-1257.R0)

See Appendix B.

RSPB-2021-1620.R0

Review form: Reviewer 1 (A. Richard Palmer)

Recommendation

Accept with minor revision (please list in comments)

Scientific importance: Is the manuscript an original and important contribution to its field?

Excellent

General interest: Is the paper of sufficient general interest?

Excellent

Quality of the paper: Is the overall quality of the paper suitable?

Acceptable

Is the length of the paper justified?

No

Should the paper be seen by a specialist statistical reviewer?

No

Do you have any concerns about statistical analyses in this paper? If so, please specify them explicitly in your report.

No

It is a condition of publication that authors make their supporting data, code and materials available - either as supplementary material or hosted in an external repository. Please rate, if applicable, the supporting data on the following criteria.

Is it accessible?

Yes

Is it clear?

Yes

Is it adequate?

Yes

Do you have any ethical concerns with this paper?

No

Comments to the Author

I have made a number of queries and minor editorial suggestions directly in the annotated PDF of the MS, the more significant ones include (line numbers as in original PDF).

2: Perhaps revise title to read "sessile acorn barnacle"

68-70: I believe all chthamaloid barnacles, and numerous balanoid barnacles (e.g., *Semibalanus*, *Membranobalanus*, *Bathylasma*, *Platylepas*) have membranous bases, all of which are considered colloquially to be "acorn barnacles", so reword this entire sentence to read "Like several other acorn barnacles, *C. testudinaria* have a flexible basal membrane, rather than a rigid calcareous one."

75-76: What is the ancestral state of the basis of acorn barnacles, calcified or membranous? If membranous (as I suspect is true), then this statement is misleading. Better to just say "possession" rather than "evolution" of a membranous basis.

77-91: Is all of this detail about barnacle mating systems needed? Could it be condensed to 2 or 3 sentences? Something like: "Barnacles might potentially move to increase the chances of mating, because even with their unusually long penises, they are nonetheless constrained to mate with nearby neighbors. This may be less important in *C. testudinaria*, where large hermaphrodites sometimes bear small, complemental males."

323: Indicate, if true, that barnacle diameters were measured at the base

353: State the approx. water velocities in this tray.

509-510: The Anderson 1994 book reference is repeated twice

589: Fig. 1- A-E, H,I- all of these panels should have scale bars in them, include photograph dates for panels D and E. In panels B, C and E, add arrowheads to the lines that connect the starting reference line to current position of the barnacle (see samples in annotated pdf), so it is more obvious which direction the barnacle was moving. Why is the circle in panel H not on top of the barnacle that moved? If there is a reason, explain it in the legend.

601: Fig. 2 title- Indicate the flow conditions of the experiment in the figure title.

628: Fig. 4 legend- State specifically what the 0 degree angle corresponds to in Panel B. Is 0 rostral or lateral? The drawing seems to indicate 0 is lateral, but the text seems to indicate that 0 means rostral.

625: First word of title of panel C needs to be capitalized; title of panel G should read "Variation in nearest neighbour index"

634: Fig. 4 legend- Insert "(see Fig. S6 for methods)"

642: Fig. 5 legend- Indicate the approx. flow velocities the barnacles experienced in these treatments.

649: Fig. 5 legend- State "attached to" rather than "laced to"?

A. Richard Palmer
(also attached as a PDF file)

Decision letter (RSPB-2021-1620.R0)

31-Aug-2021

Dear Dr Robinson:

Your manuscript has now been peer reviewed and the reviews have been assessed by an Associate Editor. The reviewers' comments (not including confidential comments to the Editor) and the comments from the Associate Editor are included at the end of this email for your reference. As you will see, the reviewers and the Editors have raised some concerns with your manuscript and we would like to invite you to revise your manuscript to address them.

Research ethics:

Use of animals and field studies:

It is a condition of publication that you make available the data and research materials supporting the results in the article (<https://royalsociety.org/journals/authors/author-guidelines/#data>). Datasets should be deposited in an appropriate publicly available repository and details of the associated accession number, link or DOI to the datasets must be included in the Data Accessibility section of the article (<https://royalsociety.org/journals/ethics-policies/data-sharing-mining/>). Reference(s) to datasets should also be included in the reference list of the article with DOIs (where available).

Please submit a copy of your revised paper within three weeks. If we do not hear from you within this time your manuscript will be rejected. If you are unable to meet this deadline please let us know as soon as possible, as we may be able to grant a short extension.

Best wishes,
Dr John Hutchinson, Editor
mailto:proceedingsb@royalsociety.org

Associate Editor
Comments to Author:

Thank you for submitting a revised version of your manuscript, addressing the points raised in the reviews of the original version. The MS has been reviewed a second time by one of the original referees, who is satisfied that the primary points identified in the original version have generally been successfully addressed. A number of specific points are noted as requiring attention in this new version, in comments provided directly on the manuscript and in a separate list. A recommendation was also made to specifically re-examine the Introduction and Discussion for opportunities to refine the text and make it more concise.

In addition to the suggestions of the Referees, I would add a few further recommended corrections that I list below.

Thank you once again for your submission. I hope you find these comments helpful in revising your report on your study.

L38. Add a comma between "and" and "in".

L99. Potentially change "in optimal positioning later as surface" to "in an ultimately optimal position, as surface..."

L102-103. Potentially change "in the field over time" to in the field, over spans of several weeks,..."

L119. Add a comma before "and".

L144. Change "was" to "were".

- L207. Discussion – citations are no longer entered in superscript.
- L213. The referee suggested change of “to increase” seems better as “increases” to be parallel with earlier portions of the sentence.
- L283. Add a comma after “currents”.
- L284. Add a comma after “feed”.
- L291. Add a comma before “which”.
- L294. Add a comma after “>10mm”.
- L321. “randomly” seems more appropriate than “haphazardly”.
- L357. Change “was repeated a second time but this time, we used” to “was repeated a second time, during which we used”.
- L376. Change “ASTM (2005)⁴⁴ and the force requires to” to “ASTM (2005)⁴⁴, and the force required to”.
- L391. Change “and thus the” to “and, thus, the”.
- L398. Add a comma after “(Fig. 5D)”.
- L409-410. Change “This experiment was repeated again but this time using” to “This experiment was repeated using”

Reviewer(s)' Comments to Author:

Referee: 1

Comments to the Author(s).

I have made a number of queries and minor editorial suggestions directly in the annotated PDF of the MS, the more significant ones include (line numbers as in original PDF).

2: Perhaps revise title to read "sessile acorn barnacle"

68-70: I believe all chthamaloid barnacles, and numerous balanoid barnacles (e.g., *Semibalanus*, *Membranobalanus*, *Bathylasma*, *Platylepas*) have membranous bases, all of which are considered colloquially to be "acorn barnacles", so reword this entire sentence to read "Like several other acorn barnacles, *C. testudinaria* have a flexible basal membrane, rather than a rigid calcareous one."

75-76: What is the ancestral state of the basis of acorn barnacles, calcified or membranous? If membranous (as I suspect is true), then this statement is misleading. Better to just say "possession" rather than "evolution" of a membranous basis.

77-91: Is all of this detail about barnacle mating systems needed? Could it be condensed to 2 or 3 sentences? Something like: "Barnacles might potentially move to increase the chances of mating, because even with their unusually long penises, they are nonetheless constrained to mate with nearby neighbors. This may be less important in *C. testudinaria*, where large hermaphrodites sometimes bear small, complementary males."

323: Indicate, if true, that barnacle diameters were measured at the base

353: State the approx. water velocities in this tray.

509-510: The Anderson 1994 book reference is repeated twice

589: Fig. 1- A-E, H,I- all of these panels should have scale bars in them, include photograph dates for panels D and E. In panels B, C and E, add arrowheads to the lines that connect the starting reference line to current position of the barnacle (see samples in annotated pdf), so it is more obvious which direction the barnacle was moving. Why is the circle in panel H not on top of the barnacle that moved? If there is a reason, explain it in the legend.

601: Fig. 2 title- Indicate the flow conditions of the experiment in the figure title.

628: Fig. 4 legend- State specifically what the 0 degree angle corresponds to in Panel B. Is 0 rostral or lateral? The drawing seems to indicate 0 is lateral, but the text seems to indicate that 0 means rostral.

625: First word of title of panel C needs to be capitalized; title of panel G should read "Variation in nearest neighbour index"

634: Fig. 4 legend- Insert "(see Fig. S6 for methods)"

642: Fig. 5 legend- Indicate the approx. flow velocities the barnacles experienced in these treatments.

649: Fig. 5 legend- State "attached to" rather than "laced to"?

A. Richard Palmer
(also attached as a PDF file)

Author's Response to Decision Letter for (RSPB-2021-1620.R0)

See Appendix C.

Decision letter (RSPB-2021-1620.R1)

13-Sep-2021

Dear Dr Robinson

I am pleased to inform you that your manuscript entitled "500 million years to mobility: Directed locomotion and its ecological function in a turtle barnacle" has been accepted for publication in Proceedings B.

Data Accessibility section

Open Access

Paper charges

Sincerely,

Proceedings B

Appendix A

Revision to Chan et al. submitted to Proceedings of the Royal Society B.

The work of Benny K.K. Chan and co-authors entitled “Confirmation of active location and its ecological function by an ostensibly sessile barnacle” presents evidence of the barnacle *Chelonibia testudinaria* is capable of motion through different substrates with predominant direction against the water flow. The authors attributed this locomotion primarily to the barnacle feeding behavior and discharged a possible reason to optimize reproduction. In addition, the authors tried to explain the mechanisms of movement by studying the cemented trails left behind by the moving barnacles. While the results appear solid and sound, the hypotheses, scientific perspective of the study should be considerably developed. There are several issues that should be addressed before considering this manuscript for publications. Please find below my comments and suggestions.

Main issues

Title

I think the title does not give the right credit to this study. I find the first part of the title misleading “Confirmation of active location”. This study confirms previous observations of active motion by barnacles with manipulative experiments. I would suggest to replace “Confirmation of” with “Evidence of...”

I would suggest to replace “active location” with “active movement” or “active displacement”, because this study shows that the barnacles move through natural and artificial substrates. The words movement and locomotion are also listed as keywords and recur in the abstract and discussion. Therefore I would rather use one of them instead of “active location”.

Introduction

Although the introduction provides sufficient information to the reader to follow the overall study, more details of the life strategy of epibiotic barnacles could be provided.

Barnacles are passive suspension feeders and in low current conditions they actively beat their cirri to optimize the feeding rates. But what about the barnacles living on turtles and other swimming organisms? Do they beat their cirri as well or this energetic expenditure is not necessary due to the water flow created by the swimming turtles? Therefore are these barnacles using energy for displacement for a better location while those fixed to a not living substrate use mostly their energy for beating cirri?

Lines 50-52: The authors mention a previous study (Moriarty et al. 2008) that reported individual translocation of *Chelonibia testudinaria* on a turtle carapace. Are there other studies reporting or suggesting movement of barnacles of other species? If not, please make it clear in the introduction and explain the reason why investigating possible active movement in barnacles is relevant from a biological and ecological prospective.

Line 60-62: I am not an expert in barnacles, and probably other readers of Proceedings B are neither too. Therefore I would suggest the authors to develop a short explanation of the two groups that are mentioned here: based on flexible membrane and rigid calcified plate. Are generally the barnacles divided in these two groups based on their plate? Or are there other species that have a different

plate? I think all this information will help the readers to better understand the findings of this study. Although this is developed in lines 202-203 of the discussions, it will help to have a short explanation in the introduction for the not expert readers.

Line 79-85: The authors do not clearly state the hypothesis/es that they want to test. They state that “Thus, we developed a suite of experiments with adult *C. testudinaria* on sea turtle, crab, and synthetic substrata to confirm active directional translocation in this species.”

The hypothesis and objectives of this study should be clearly stated and developed at the end of the introduction.

Results

Line 124-133: The authors estimated the detachment force average after different re-attached timespans and different substrate (artificial and natural). They found statistical significant differences between the artificial and natural substratum, although this is not observed after 365 days of re-attachment.

However, the area range changes over the reattachment time (days): for the artificial $\leq 1.2 \text{ cm}^2$ and only after 365 days it reached a bigger size, whereas for natural is between 0.9 and 5 cm^2 .

I understand that the area range, that reflects the shell diameter or cementation area, can be considered as the surface area attached to the substratum.

From the data shown in figure S5 is possible to observed that the detachment force increases with the increases of area range. Have the authors tested whether the area range might be a factor that affect the force at which the barnacle is attached. I mean, as much as bigger is the shell, bigger is the area attached to the substrate and as a consequence the force to detach the barnacle should be stronger.

Line 140-143: In the figure 3 the authors indicate two different types of cement, heterogeneous and homogeneous, and how they alternate as long as the barnacle moves. In the result text this phenomenon is not fully described. The authors mention a “bi-layered construction” in line 142, but they do not properly describe it. I would suggest to add a sentence implementing the description of such “bi-layered construction” as well as describe heterogeneous and homogenous cement structure shown in the figure 3D and E.

Line 158: The authors demonstrated that when pairs of barnacles were reattached to artificial substratum, the distance between the barnacles did not change over the time. However, later they stated, “Individuals within pairs appear to move randomly”. This sentence is not supported by any data/figure and it sounds contractive to what they stated before.

Could you please provide a more developed description of what you want to communicate with this sentence together with proper results description?

Figure 4D is shown, but the results are not explained in the main text. All the shown results should be explained in the main text.

Discussion

The discussion is clear and straight to the point. However, as long as I read it, I would have appreciated further details and some questions came up to my mind. The authors should stress out the results in an ecological context: are other species of barnacles observed or suggested to be able to move? If some species are able to move and others might not be able, could have different species evolved a different behavior and therefore a different feeding strategy? Could be the active movement a common phenomenon in epibiotic species or also in species attached to rocky substrate? This study demonstrated that the barnacles moved facing the current, which represent the direction of swimming. But what about the barnacles attached to a rocky substrate?

Line 221-225: The authors rule out the movement as a consequence to optimize reproduction. Is there any reproduction season in barnacles as in other invertebrates? If this is the case, maybe the lack of movement–reproduction forces could have been masked by the fact that the experiments were carried out during not reproduction season?

It might be that they avoid to stay closer, because the close vicinity might be disadvantageous for feeding?

It would be interesting the authors address these questions in the text.

Line 234-235: This is very interesting. When there is not a unidirectional current flow it seems that the barnacles wander. Unfortunately, the authors do not provide any suggestion of this random movement. Why do you think is the reason the barnacles move anyway? Is that a losing of energetic expenditure? From an energetic point of view, I would have expect the barnacles not moving and invest energy in beating their cirri.

Methods

Line 258-266: The authors explain that the turtles used in this experiments were incidentally caught by local fisheries in Spain and then brought to a recuperation and conservation center. However the authors do not explain how the turtles were kept and used for this experiment. This information should be added in the main text, because it will be not only useful for an experimental point of view but also ethical as *Carretta caretta* is considered an endangered species.

Line 277: please define “medial direction”. In general to avoid misunderstanding I would suggest to define each of the movements listed here. I assume that the movement is related to the direction of the barnacle respect to the turtle carapace. For example, I assume that “anterior” means going towards the turtle head, “posterior” toward the back of the turtle, “lateral” towards the edge of the turtle carapace, but what about medial?

Minor issues:

Keywords

I would suggest to add “barnacles” as key word.

Introduction

Line 76: Please replace “highest” with “higher”.

Abstract

Line 39: Please replace the adjective “independent” with the adverb “independently”.

Results

In general, I would suggest to use the same font size for the panel letters “a”, “b”, “c” throughout the text.

Line 95: What does “medial direction” mean? Please define it.

Line 101: Please replace “to studies” with infinitive “to study”

Line 106: Figure S2 comes before figure S1 in the main text.

Line 152: Please replace “all” at the beginning of the sentence, so it reads as “ All barnacles in the control treatment moved distances ...”

Figure 2 caption: The panel F is mentioned in the figure caption, but no figure 2F is shown.

Figure 2E: The axis labels and numbers are too small to be readable. I would suggest to increase the letter size.

Line 155: Section title “Does locomotion improve reproductive conditions?” Please use the same font size for all title subsections.

Methods

Line 341-343: Please write “Barnacles” with lowercase letter “barnacles”. There is no need to use capita letters here.

Discussion

Line 218: If adult *C. testudinaria* are uniquely capable.....

Please replace “adult” with plural “adults” as the verb refers to plural, so the sentence read as “If *C. testudinaria* adults are uniquely capable...”

Line 348_ the entire species name should be written in italic.

Supplementary materials

Figure S1: It is not clear to me what the Y axis shows. Is it the distance travelled normalized by shell size or the two parameters (shell diameter and distance travelled) are shown separately? If the latter is the case, I would suggest to add an extra Y axis, maybe on the right side of the panel.

Furthermore Figure S2 comes before Figure S1 in the main text. Please use sequential number.

Figure S4: The figure legend is too small to be readable. Although the different colors are explained in the figure caption, I would suggest to make the figure legends bigger so this plot is easier to read.

Figure S6: I find this figure very informative of the overall experimental procedure. I would suggest to replace this figure in the main text, if this applies with the journal guidelines.

Appendix B

Dear Editor

Re: Revised manuscript

Thank you for your email dated 29 June 2021 informing us of the opportunity to address the comments of reviewers on our manuscript and for the possibility to submit a new version. We have modified the manuscript and addressed all the reviewers' comments with our specific responses detailed below:

Reviewer 1:

Comment: Title: I think the title does not give the right credit to this study. I find the first part of the title misleading "Confirmation of active location". This study confirms previous observations of active motion by barnacles with manipulative experiments. I would suggest to replace "Confirmation of" with "Evidence of..."

I would suggest to replace "active location" with "active movement" or "active displacement", because this study shows that the barnacles move through natural and artificial substrates. The words movement and locomotion are also listed as keywords and recur in the abstract and discussion. Therefore I would rather use one of them instead of "active location".

Response: Thanks for these suggestions on the title. We have reframed it to the following: "500 million years to mobility: Directed translocation and its ecological function in an ostensibly sessile barnacle". This title removes the word "confirmation" and more clearly articulates the subject of the MS.

Comment: *Introduction.* Although the introduction provides sufficient information to the reader to follow the overall study, more details of the life strategy of epibiotic barnacles could be provided.

Response: In the revised introduction, we have added more about what is known of movement in barnacles generally and elaborated on the life strategy of *C. testudinaria*. Lines 52-60. "Once affixed to a substratum by their adhesive cement⁶⁻¹⁰, thoracican barnacles are generally considered irreversibly attached and incapable of further movement, though some species can make small adjustments in their point of attachment. Acorn barnacles (Balanomorpha) can shift position by asymmetric growth and shell repair¹¹ and gooseneck species (Scalpellomorpha) through growth in combination with contractions of the muscular stalk^{12,13}. There is also evidence that some barnacles can reattach when partially dislodged¹⁴⁻¹⁶. However, in striking contrast to these minor movements, the epizoic barnacle *Chelonibia testudinaria* (Linnaeus, 1758) is reported translocating distances > 30 cm over several months' time across the shells of sea turtles¹⁷."

Comment: Barnacles are passive suspension feeders and in low current conditions they actively beat their cirri to optimize the feeding rates. But what about the barnacles living on turtles and other swimming organisms? Do they beat their cirri as well or this energetic expenditure is not necessary due to the water flow created by the swimming turtles? Therefore are these barnacles using energy for displacement for a better location while those fixed to a not living substrate use mostly their energy for beating cirri?

Response: The reviewer makes an excellent point. There are two general types of cirral activities in barnacles. Many species actively beat the cirri for capturing plankton in the absence of flow but feed passively in flow. We have elaborated on this as well as a commenting on the extreme passivity we have observed in *Chelonibia*, a statement we now support with reference to our unpublished data. Lines 100-104 "Barnacles passively suspension feed in current on plankton and organic debris by means of a fan of cirral appendages. When flow is lacking, balanid species are known to actively beat their cirri, an energetically expensive endeavour, to optimize feeding rates³¹. *Chelonibia testudinaria* on the other hand is remarkably passive, spending little to no time actively

feeding even in prolonged absences of flow (Zardus, unpublished data).”

Comment: Lines 50-52: The authors mention a previous study (Moriarty et al. 2008) that reported individual translocation of *Chelonibia testudinaria* on a turtle carapace. Are there other studies reporting or suggesting movement of barnacles of other species? If not, please make it clear in the introduction and explain the reason why investigating possible active movement in barnacles is relevant from a biological and ecological prospective.

Response: Moriarty et al. is the only study reporting that this or any barnacle is able to move at the scale observed. We have revised the introduction to provide more information on what is known about barnacle movement generally and how very minimal that movement is in every other species known. The possibility that movement in *Chelonibia* involves behaviourally directed locomotion is fascinating considering that adult barnacles presumably lost the capacity for active movement when they first evolved 500 million years ago (as now highlighted in our title). We have also made more explicit the connection between movement in *Chelonibia* and the paradigm of movement ecology which seeks to answer how, where, when, and why organisms move by addressing these four elements throughout the text. We feel this barnacle represents a novel application of this body of theory.

Comment: Line 60-62: I am not an expert in barnacles, and probably other readers of Proceedings B are neither too. Therefore I would suggest the authors to develop a short explanation of the two groups that are mentioned here: based on flexible membrane and rigid calcified plate. Are generally the barnacles divided in these two groups based on their plate? Or are there other species that have a different plate? I think all this information will help the readers to better understand the findings of this study. Although this is developed in lines 202-203 of the discussions, it will help to have a short explanation in the introduction for the not expert readers.

Response: We have more clearly introduced the differences and consequences of the two types of bases in barnacles (rigid calcareous base vs flexible membranous). Attachment properties vary with the two bases and we now highlight that a membranous base may have been an evolutionary pre-adaptation for movement. Lines 75-83 “Of these two types of bases in acorn barnacles, those that are calcified are rigid, impermeable, and cemented to the substratum. Basal plates are secreted by a membrane invested with ovarian tubules and cement glands and the surrounding conical shell interfaces with the base around its perimeter²⁰. Shells with basal plates do not easily detach, even when the barnacle is dead, and when they do so the plate is often left behind. In comparison, membranous bases are entirely organic, and when barnacles with this type of base die the shell detaches from the substratum and any remaining membranous tissue decomposes²¹. The evolution of a flexible, non-calcified base was likely a critical pre-adaptation for movement in this species.”

Comment: Line 79-85: The authors do not clearly state the hypothesis/es that they want to test. They state that “Thus, we developed a suite of experiments with adult *C. testudinaria* on sea turtle, crab, and synthetic substrata to confirm active directional translocation in this species.”

The hypothesis and objectives of this study should be clearly stated and developed at the end of the introduction.

Response: We have revised the last paragraph of the introduction and added in the two hypotheses we are testing, that movement enhances 1) feeding or 2) reproduction. Lines 109-114 “To confirm the extent of mobility in *C. testudinaria*, we carried out observations in the field over time of barnacles on wild and captive sea turtles. We also conducted a suite of laboratory experiments using individual barnacles reattached to synthetic substrata to test the hypotheses that their movement functions to enhance feeding or mating. Furthermore, we examined the barnacle adhesive cement by light and

Scanning Electron Microscopy (SEM) for insight on the mechanisms of movement and mechanics of adhesion.”

Comment: *Results.* Line 124- 133: The authors estimated the detachment force average after different re- attached timespans and different substrate (artificial and natural).

They found statistical significant differences between the artificial and natural substratum, although this is not observed after 365 days of re- attachment.

However, the area range changes over the reattachment time (days): for the artificial ≤ 1.2 cm² and only after 365 days it reached a bigger size, whereas for natural is between 0.9 and 5 cm².

I understand that the area range, that reflects the shell diameter or cementation area, can be considered as the surface area attached to the substratum.

From the data shown in figure S5 is possible to observed that the detachment force increases with the increases of area range. Have the authors tested whether the area range might be a factor that affect the force at which the barnacle is attached. I mean, as much as bigger is the shell, bigger is the area attached to the substrate and as a consequence the force to detach the barnacle should be stronger.

Response: The reviewer’s point is valid, as barnacles increase in size their attachment strength may increase simply to an increase in surface area or amount of glue. However, our objective for this measurement is to demonstrate that the barnacles were attached strongly enough from an early period of re-attachment, so excluding the possibility of translocation by external forces. We would rather not digress into detailed analysis of changes in attachment strength over time, especially since our measurements between reattached and naturally attached individuals of similar size were not significantly different. We have now added language that refers to the likely effect of size on adhesion strength and points out the important results comparing reattached and naturally attached individuals of similar size. Lines 164-169 “Smaller, newly re-attached specimens required less force to remove than the larger, naturally attached individuals on crabs ($F_{4, 30} = 4197$, $p < 0.05$), likely a consequence of the difference in size and surface area available for adhesion. However, post-hoc SNK tests revealed that as the barnacles grew their adhesive strength increased over time in re-attached individuals (Fig. S5) and that by 365 days the forces needed to remove individuals on synthetic panels (~42N) were not statistically different from specimens of similar size on crabs (~48N) (Fig. S5).”

Comment: Line 140- 143: In the figure 3 the authors indicate two different types of cement, heterogeneous and homogeneous, and how they alternate as long as the barnacle moves. In the result text this phenomenon is not fully described. The authors mention a “bi-layered construction” in line 142, but they do not properly describe it. I would suggest to add a sentence implementing the description of such “bi-layered construction” as well as describe heterogeneous and homogenous cement structure shown in the figure 3D and E.

Response: We have revised and elaborated on the bi-layered construction. In lines 176-184, we now state “Further investigation of the cement trails under SEM (Fig. 3A-E) revealed microstructural variations that coincided with position (time since secretion). When viewed from the side, newer cement appeared homogeneous with cubic crystals and originated from below adjacent, older deposits which appeared heterogeneous (Fig. 3B, C), while each instalment exhibited an alternating bi-layered construction indicating some discontinuity during secretion (Fig. 3). Appearing in regular, alternating arrangement (Fig. 3F), the outer layer looked similar to secondary cement seen in barnacles that have partially lifted and reattached. Such cement trails further supported the conclusion that secretion and movement was incremental and not continuous (Fig. 3F).

Comment: Line 158: The authors demonstrated that when pairs of barnacles were

reattached to artificial substratum, the distance between the barnacles did not change over the time. However, later they stated, “Individuals within pairs appear to move randomly”. This sentence is not supported by any data/figure and it sounds contractive to what they stated before.

Could you please provide a more developed description of what you want to communicate with this sentence together with proper results description?

Response: We have clarified this statement in lines 197-202 with “When pairs of barnacles were laced on to vertically positioned acrylic plates at three inter-individual treatment distances (50 mm, 100 mm, and 150 mm), there was no consistent movement observed shortening inter-neighbour distances within treatments (Fig. 4C). Individuals within pairs moved in all directions, with most individuals repositioning themselves at angles of 180-200° (Fig. 4D) and there was no significant change in the initial and final distances between the barnacles (paired t-test, $p > 0.05$; Fig. 4C).

Comment: Figure 4D is shown, but the results are not explained in the main text. All the shown results should be explained in the main text.

Response: This is now addressed in line 201 (see the response in the previous comment).

Comment: *Discussion.* The discussion is clear and straight to the point. However, as long as I read it, I would have appreciated further details and some questions came up to my mind.

The authors should stress out the results in an ecological context: are other species of barnacles observed or suggested to be able to move? If some species are able to move and others might not be able, could have different species evolved a different behavior and therefore a different feeding strategy? Could be the active movement a common phenomenon in epibiotic species or also in species attached to rocky substrate? This study demonstrated that the barnacles moved facing the current, which represent the direction of swimming. But what about the barnacles attached to a rocky substrate?

Response: We suggest that calcareous-based species cannot move. In lines 301-311, we now state “The mechanism enabling these animals to translocate requires further study, but the ecological impact of this adaptation is undoubtedly profound for *C. testudinaria*. Other barnacle species that are also epizoic with sea turtles^{34,40} likely face similar pressures with regards to optimal positioning for filter-feeding. However, none attach in precisely the same manner as *C. testudinaria*, a factor that is likely significant to its movement. For intertidal or rocky substrate barnacles, some have rigid calcareous basal plates that probably inhibit their capacity to evolve locomotion. But for those with a membranous base, movement by *C. testudinaria* raises interest in determining whether locomotion in similar barnacles is possible or in other ‘sessile’ species. Indeed, a recent study has found that some deep-sea sponges are also unexpectedly capable of meandering spatial translocation⁴¹, providing additional incentive to study the selective pressures in the movement ecology of sessile invertebrates generally.”

Comment: Line 221-225: The authors rule out the movement as a consequence to optimize reproduction. Is there any reproduction season in barnacles as in other invertebrates? If this is the case, maybe the lack of movement–reproduction forces could have been masked by the fact that the experiments were carried out during not reproduction season?

It might be that they avoid to stay closer, because the close vicinity might be disadvantageous for feeding?

It would be interesting the authors address these questions in the text.

Response: Reproduction in barnacles is strongly temperature dependent. We kept the water temperature in the movement experiment at 25°C which is the summer temperature of seawater in Taiwan and kept the animals well-fed. Dissection of dead individuals in the experiment revealed the individuals had mature gonads. This has now been addressed in lines 442-446 “During all laboratory movement experiments, barnacles were kept at 25 °C

water temperature, which is typical of seawater temperatures during summer in Taiwan when most barnacles produce mature gonads for reproduction. *Chelonibia testudinaria* had mature ovary and testis upon dissection of dead individuals in the experiments, suggesting that individuals were reproductively active during all movement experiments.”

Comment: Line 234- 235: This is very interesting. When there is not a unidirectional current flow it seems that the barnacles wander. Unfortunately, the authors do not provide any suggestion of this random movement. Why do you think is the reason the barnacles move anyway? Is that a losing of energetic expenditure? From an energetic point of view, I would have expect the barnacles not moving and invest energy in beating their cirri.

Response: This is a great question. Without stimulation from water currents, barnacles may move randomly in search of better feeding conditions. As mentioned above, cirral acitivity in *C. testudinaria* is extremely passive, which may inspire the need to move to a better location. Without further studies on the energetics it is difficult to say if the cost for active feeding is higher than locomotion. We have followed up on the query by including the following, lines 290-298 “If improved feeding position is the primary gain for locomotion in barnacles, then their movements may be continuous and perennial rather than cyclical. On the other hand, it is possible that where and when barnacles move varies in direction and intensity over time and is perhaps linked to barnacle life-history or turtle size or age class. In the absence of flow it was interesting to note that barnacles moved in random directions, presumably in search of feeding currents. Lacking information on energetics, it would be interesting to know if there is a trade-off in locomotion versus active cirral beating. The predominance of passive feeding in *C. testudinaria* may indicate that movement is less costly than staying fixed in a location where extensive active feeding is necessary.”

Comment: *Methods.* Line 258- 266: The authors explain that the turtles used in this experiments were incidentally caught by local fisheries in Spain and then brought to a recuperation and conservation center. However the authors do not explain how the turtles werekept and used for this experiment. This information should be added in the main text, because it will be not only useful for an experimental point of view but also ethical as *Carretta caretta* is considered an endangered species.

Response: We have added several sentences in the methods section to describe both the permits and the animal care received by the turtles during rehabilitation and included a citation to the details of veterinary care (Arkwright et al. 2020), line 326-331 “Upon admission to ARCA, each of these animals was suffering from decompression sickness and was thus treated and monitored for up to 18 weeks before being released back into the wild (for details on veterinary care, see ⁴²). Sea turtles were housed in circular tanks, ranging from 2 to 6 m in diameter with a water depth of 0.95 m and maintained at a water temperature of ~24°C. All animals were fed twice daily using a mix of vegetable and fish material.”.

Comment: Line 277: please define “medial direction”. In general to avoid misunderstanding I would suggest to define each of the movements listed here. I assume that themovement is related to the direction of the barnacle respect to the turtle carapace. For example, I assume that “anterior” means going towards the turtle head, “posterior” toward the back of the turtle, “lateral” towards the edge of the turtle carapace, but what about medial?

Response:

We have tried to improve the clarity of the text and now define what we mean by the different directions of movements when we first reference this on lines 127-130. This section now reads “We observed that 14 of the barnacles headed towards the front of the carapace (i.e. anteriorly), 9 traveled to the back to the carapace (i.e. posteriorly), 5 wandered towards the mid-line of the carapace (i.e. medially), while no barnacles moved away from the mid-line of the carapace (i.e. laterally) (Fig. 1F).”

Comment: Minor issues:

Keywords

I would suggest to add “barnacles” as key word.

Response: added.

Comment: Editing comments:

Introduction Line 76: Please replace “highest” with “higher”.

Abstract Line 39: Please replace the adjective “independent” with the adverb “independently”.

Results In general, I would suggest to use the same font size for the panel letters “a”, “b”, “c” throughout the text.

Line 101: Please replace “to studies” with infinitive “to study”

Line 152: Please replace “all” at the beginning of the sentence, so it reads as “ All barnacles in the control treatment moved distances ...”

Line 155: Section title “Does locomotion improve reproductive conditions?” Please use the same font size for all title subsections.

Methods

Line 341- 343: Please write “Barnacles” with lowercase letter “barnacles”. There is no need to use capita letters here.

Discussion

Line 218: If adult *C. testudinaria* are uniquely capable.....

Please replace “adult” with plural “adults” as the verb refers to plural, so the sentence read as “If *C. testudinaria* adults are uniquely capable...”

Line 348_ the entire species name should be written in italic.

Response: All editing comments have been addressed.

Comment: Line 95: What does “medial direction” mean? Please define it.

Response: We have clarified the text and removed the redundant percentages that are available in the figures. This section now reads. Lines 127-130 “We observed that 14 of the barnacles headed towards the front of the carapace (i.e. anteriorly), 9 traveled to the back to the carapace (i.e. posteriorly), 5 wandered towards the mid-line of the carapace (i.e. medially), while no barnacles moved away from the mid-line of the carapace (i.e. laterally) (Fig. 1F).”

Comment: Line 106: Figure S2 comes before figure S1 in the main text.

Response: In lines 112-132, we now cited Figure S1, so it comes before S2.

Comment: Figure 2 caption: The panel F is mentioned in the figure caption, but no figure 2F is shown.

Response: Figure 2F caption has been removed.

Comment: Figure 2E: The axis labels and numbers are too small to be readable. I would suggest to increase the letter size.

Response: All fonts and labels have been enlarged.

Comment: *Supplementary materials*

Figure S1: It is not clear to me what the Y axis shows. Is it the distance travelled normalized by shell size or the two parameters (shell diameter and distance travelled) are shown separately? If the latter is the case, I would suggest to add an extra Y axis, maybe on the right side of the panel.

Response: In the original graph, we used the same axis to show both growth in terms of

barnacle size and the distance travelled as they are both measured in the same units (mm). We have incorporated the feedback by adding a second axis.

Comment: Furthermore Figure S2 comes before Figure S1 in the main text. Please use sequential number.

Response: It is now corrected to follow the sequences of Figure S1 and then S2.

Comment: Figure S4: The figure legend is too small to be readable. Although the different colors are explained in the figure caption, I would suggest to make the figure legends bigger so this plot is easier to read.

Response: The legend has been enlarged.

Comment: Figure S6: I find this figure very informative of the overall experimental procedure. I would suggest to replace this figure in the main text, if this applies with the journal guidelines.

Response: Figure S6 is now included as Figure 6 in the MS.

Reviewer 2:

All editing comments marked on the pdf have been addressed.

Comment: The % barnacles moving in given directions in the text does not agree with the numbers in Fig. 1f. For example, the text says "(47 %) headed towards the anterior of the turtles' carapace", but Fig. 1F indicates only 27-28% did. These discrepancies should be fixed.

Response: Both numbers are correct. Figure 1F shows the percentage of barnacles that moved in each direction considering both moving and non-moving barnacles (e.g. all 50 barnacles). However, we have amended the text to avoid any confusion by removing the percentages and leaving only the numbers of moving barnacles. Line 127-130, "We observed that 14 of the barnacles headed towards the front of the carapace (i.e. anteriorly), 9 traveled to the back to the carapace (i.e. posteriorly), 5 wandered towards the mid-line of the carapace (i.e. medially), while no barnacles moved away from the mid-line of the carapace (i.e. laterally) (Fig. 1F)"

Comment: Comment on approx. water velocities in the aquarium and how barnacles were positioned relative to the direction of flow.

Response: The flow rate of the pump is 60l hr⁻¹, with the diameter of outflow tube is 9.24 mm, we calculated the flow velocity as 2.48 ms⁻¹. We have addressed this in lines 408-410, "The internal diameter of the outflow tube of the underwater pump was 9.24 mm and thus the flow velocity was 2.48 m s⁻¹ (Velocity = flow rate x tube cross section area)."

Comment: state explicitly what the direction of the "detachment force" was: a) shear (parallel to the substratum, as in ASTM 2011 (ref. 36)? or b) perpendicular to the substratum. If the latter, the authors should indicate how they attached the force gauge to the barnacle prior to detachment.

Response: We used a parallel shear force. We have revised the content as follows, lines 391-394 "To compare the attachment strength among barnacles that were either experimentally reattached to acrylic plates or naturally attached to crab carapaces, a shear force was applied to each test individual parallel to the substratum following methods specified in ASTM (2005)⁴⁴ and the force requires to detach them was measured with a force gauge (FG-20G, Taiwan)."

Appendix C

Response to reviewer comments:

Associate Editor

Comments to Author:

Thank you for submitting a revised version of your manuscript, addressing the points raised in the reviews of the original version. The MS has been reviewed a second time by one of the original referees, who is satisfied that the primary points identified in the original version have generally been successfully addressed. A number of specific points are noted as requiring attention in this new version, in comments provided directly on the manuscript and in a separate list. A recommendation was also made to specifically re-examine the Introduction and Discussion for opportunities to refine the text and make it more concise.

In addition to the suggestions of the Referees, I would add a few further recommended corrections that I list below.

Thank you once again for your submission. I hope you find these comments helpful in revising your report on your study.

Response:

Dear editor,

We sincerely thank you and the reviewer for going over our manuscript and paying attention to small yet crucial details. We have re-examined the introduction and discussion and made it more concise and clear following the suggestions by the reviewer. We hope our re-submission is satisfactory for publication in your journal.

On behalf of all authors,
Benny KK Chan, corresponding author

Comment: L38. Add a comma between “and” and “in”.

Response: Amended.

Comment: L99. Potentially change “in optimal positioning later as surface” to “in an ultimately optimal position, as surface...”

Response: Amended.

Comment: L102-103. Potentially change “in the field over time” to in the field, over spans of several weeks,...”

Response: Amended.

Comment: L119. Add a comma before “and”.

Response: Amended.

Comment: L144. Change “was” to “were”.

Response: Amended.

Comment: L207. Discussion – citations are no longer entered in superscript.

Response: Amended.

Comment: L213. The referee suggested change of “to increase” seems better as “increases” to be parallel with earlier portions of the sentence.

Response: Amended.

Comment: L283. Add a comma after “currents”.

Response: Amended.

Comment: L284. Add a comma after “feed”.

Response: Amended.

Comment: L291. Add a comma before “which”.

Response: Amended.

Comment: L294. Add a comma after “>10mm”.

Response: Amended.

Comment: L321. “randomly” seems more appropriate than “haphazardly”.

Response: Amended.

Comment: L357. Change “was repeated a second time but this time, we used” to “was repeated a second time, during which we used”.

Response: Amended.

Comment: L376. Change “ASTM (2005)44 and the force requires to” to “ASTM (2005)44, and the force required to”.

Response: Amended.

Comment: L391. Change “and thus the” to “and, thus, the”.

Response: Amended.

Comment: L398. Add a comma after “(Fig. 5D)”.

Response: Amended.

Comment: L409-410. Change “This experiment was repeated again but this time using” to “This experiment was repeated using”

Response: Amended.

Reviewer(s)' Comments to Author:

Referee: 1

Comments to the Author(s).

I have made a number of queries and minor editorial suggestions directly in the annotated PDF of the MS, the more significant ones include (line numbers as in original PDF).

Response: We first thank Dr. Palmer for reading our manuscript so thoroughly and for improving language and grammar. All comments and editorial suggestions from the PDF file have been amended in the manuscript, including a revision of the sentence on the sexual ecology and androdioecious mating system in the discussion.

Comment: 2: Perhaps revise title to read "sessile acorn barnacle"

Response: All barnacles are sessile so our initial wording was not optimal. We have revised the title, which is now: "*500 million years to mobility: Directed locomotion and its ecological function in a turtle barnacle*". We used turtle barnacle as this species is commonly known as the "turtle barnacle" even though it has multiple hosts and this indicates towards the epizotic nature of this barnacle.

Comment: 68-70: I believe all chthamaloid barnacles, and numerous balanoid barnacles (e.g., *Semibalanus*, *Membranobalanus*, *Bathylasma*, *Platylepas*) have membranous bases, all of which are considered colloquially to be "acorn barnacles", so reword this entire sentence to read "Like several other acorn barnacles, *C. testudinaria* have a flexible basal membrane, rather than a rigid calcareous one."

Response: This is correct, and we appreciate the attention to detail. Amended. The new L70-73 states: "To explain how *C. testudinaria* might be capable to active locomotion, it must be noted that the basal surface of this barnacle, like several other acorn barnacles, is a flexible membrane rather than a stiff calcified plate as seen in the fouling model species *Amphibalanus amphitrite*."

Comment: 75-76: What is the ancestral state of the basis of acorn barnacles, calcified or membranous? If membranous (as I suspect is true), then this statement is misleading. Better to just say "possession" rather than "evolution" of a membranous basis.

Response. This is an interesting project, which has not yet been assessed integratively with molecular and fossil data. Following the cladistic analyses provided by our group recently (Chan et al. 2021 Zool J Linn Soc), which included fossil taxa, support is giving for a calcified base being ancestral. The Brachylepadidae, the stem group of Verrucomorpha (exemplified by *Eoverruca hewittii*) and the Pachydiademmatidae all exhibit calcified bases. However, the phylogenetic analyses of recent taxa by Pérez-Losada et al. (2012 Mol Phylogenet Evol) shows that a membraenous base is the ancestral state of

the Verrucomoprha+Balanomorpha clade. This leaves the question still-open for debate. We take a conservative and cautious approach and follow the reviewers suggestion and write “possession” rather than “evolution”. The new sentence on L79-80 states: “*The possession of a flexible, non-calcified base has likely been a critical pre-adaptation for movement in this species.*”

Comment: 77-91: Is all of this detail about barnacle mating systems needed? Could it be condensed to 2 or 3 sentences? Something like: "Barnacles might potentially move to increase the chances of mating, because even with their unusually long penises, they are nonetheless constrained to mate with nearby neighbors. This may be less important in *C. testudinaria*, where large hermaphrodites sometimes bear small, complemental males."

Response: It is a complex topic and we feel some context is needed for outsiders. We take the reviewers suggestion and optimized the sentence. The new L84-88 states: “*Barnacles may potentially move to increase chances of mating as they are, even with their unusually long penises, constrained to mating with adjacent neighbors*^{24,25}. *C. testudinaria* sport an exceedingly rare sexual system in which dwarf males and hermaphrodites occur within the same reproductive population²⁶⁻³⁰ (i.e., androdioecy). Thus, movement may be less critical for ensuring mating in this species.”

Comment: 323: Indicate, if true, that barnacle diameters were measured at the base

Response: It is basal diameter, revised in the MS.

Comment: 353: State the approx. water velocities in this tray.

In this experiment, the barnacles were kept in a closed tank with aeration only. It is not possible to calculate the water velocity in the tank.

Comment: 509-510: The Anderson 1994 book reference is repeated twice

Response: Amended.

Comment: 589: Fig. 1- A-E, H,I- all of these panels should have scale bars in them, include photograph dates for panels D and E. In panels B, C and E, add arrowheads to the lines that connect the starting reference line to current position of the barnacle (see samples in annotated pdf), so it is more obvious which direction the barnacle was moving. Why is the circle in panel H not on top of the barnacle that moved? If there is a reason, explain it in the legend.

Response: We disagree that scale bars are needed in these images. Scale bars are appropriate for 2D images but not 3D images such as this. Furthermore, the turtle’s carapace can function as its own scale bar in this instance. As such, we included the Curved Carapace Length of the turtle in the image in the legend. We added dates to figure 1D and 1E.

Comment: 601: Fig. 2 title- Indicate the flow conditions of the experiment in the figure title.

Response: Barnacles in this experiment were kept in tanks with aeration only. This has now mentioned in the legend of Figure 2.

Comment: 628: Fig. 4 legend- State specifically what the 0 degree angle corresponds to in Panel B. Is 0 rostral or lateral? The drawing seems to indicate 0 is lateral, but the text seems to indicate that 0 means rostral.

Response: We adapted the figures to indicate that 0 degrees was the direction of the current flow. As such as movement of 0 degrees would equal moving directly into the current. We think this will make it easier to follow.

Comment: 625: First word of title of panel C needs to be capitalized; title of panel G should read "Variation in nearest neighbour index"

Response: Addressed

Comment: 634: Fig. 4 legend- Insert "(see Fig. S6 for methods)"

Response: Figure S6 become Fig. 5 in the revised MS. We have addressed this in the revised legend.

Comment: 642: Fig. 5 legend- Indicate the approx. flow velocities the barnacles experienced in these treatments.

Response: Revised and indicated in the revised legend

Comment: 649: Fig. 5 legend- State "attached to" rather than "laced to"?

Response: It is laced.